# REPRESENTATIVE GUIDANCE: DIFFUSION SAMPLING WITH COHERENCE

**Anh-Dung Dinh**
School of Computer Science
The University of Sydney
anh-dung.dinh@sydney.edu.au

**Daochang Liu**
School of Physics, Mathematics and Computing
The University of Western Australia
daochang.liu@uwa.edu.au

**Chang Xu** *
School of Computer Science
The University of Sydney
c.xu@sydney.edu.au

## ABSTRACT

The diffusion sampling process faces a persistent challenge stemming from its incoherence, attributable to varying noise directions across different timesteps. Our Representative Guidance (RepG) offers a new perspective to address this issue by reformulating the sampling process with a coherent direction toward a representative target. From this perspective, classic classifier guidance reveals its drawback in lacking meaningful representative information, as the features it relies on are optimized for discrimination and tend to highlight only a narrow set of class-specific cues. This focus often sacrifices diversity and increases the risk of adversarial generation. In contrast, we leverage self-supervised representations as the coherent target and treat sampling as a downstream task—one that focuses on refining image details and correcting generation errors, rather than settling for oversimplified outputs. Our Representative Guidance achieves superior performance and demonstrates the potential of pre-trained self-supervised models in guiding diffusion sampling. Our findings show that RepG not only significantly improves vanilla diffusion sampling, but also surpasses state-of-the-art benchmarks when combined with classifier-free guidance. source code: https://github.com/dungdinhanh/rep-guidance.

## 1 INTRODUCTION

In diffusion sampling processes Ho et al. (2020), a persistent challenge arises from incoherence due to uncontrollable noise introduced at each timestep. As illustrated in Figure 1, at each timestep, $\mathbf{x}_t$ is used to predict the original image, which then aids in generating $\mathbf{x}_{t-1}$ in the next step. During training, the original images are sampled from the dataset, ensuring a consistent image distribution across all timesteps. However, during inference, the real dataset distribution is unavailable. Instead, the diffusion model draws from varying distributions at each timestep, incorporating different types of information, as shown in the bottom row of Figure 1. This distributional shift between timesteps introduces incoherent features into the generated images. This paper addresses incoherence by framing it as a discrepancy between the predicted image distributions across successive timesteps. Such discrepancies allow noise information to persist, leading to undesired artifacts in the generated images. For instance, an image of a Leonberger may exhibit bizarre or inconsistent features in consecutive timesteps, hindering its transformation into a realistic depiction as the sampling process progresses, as illustrated in Figure 2. Moreover, the generated images often lack crucial details, such as background elements and fine object features. While efforts such as DDIM Song et al. (2020a) have attempted to alleviate incoherence by removing random noise during sampling, they often do so at the cost of sample quality. As a result, many recent diffusion models continue to rely on the mechanisms of conventional DDPMs Ho et al. (2020).

---

*Corresponding Author

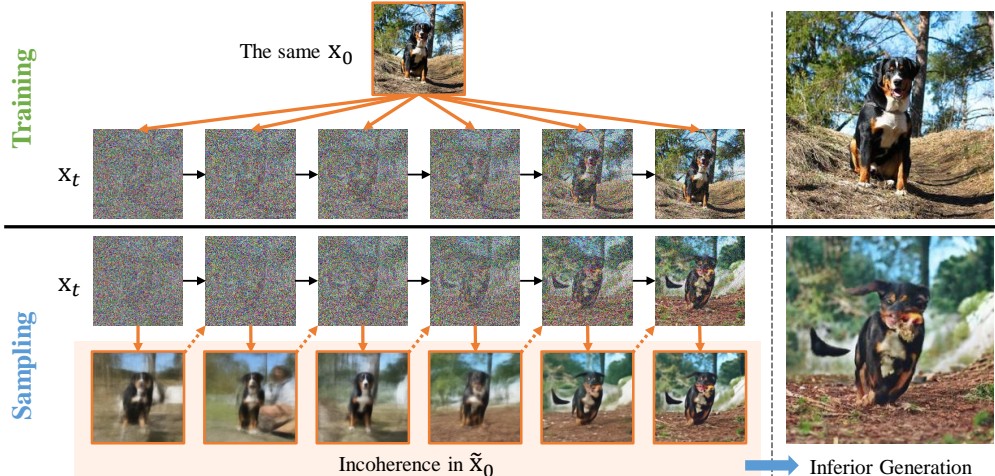

Figure 1: *The top row is the real sampling process during training, where at every timestep, real images are picked from a coherent distribution. Nevertheless, during the inference phase, as in the row below, the predicted images at every time step have different distributions. The images with earlier timesteps are more blurred than the images in the later stage of the sampling. This results in the incoherence between intermediate distributions.*

Under our formulation of the incoherence, we propose a solution that involves tuning image features at each timestep to rectify incoherent features. We introduce a guidance scheme termed Representative Guidance (RepG), which leverages information from representative vectors to steer the sampling process towards a coherent direction. Moreover, unlike the traditional classifier guidance, where one-hot vectors represent classes, RepG represents each class through a set of representative vectors containing features specific to that class. To harness the optimal representative information, we employ self-supervised models prevalent in representative learning as our guidance model. The gradients derived using the pre-trained self-supervised model are directly integrated into the sampling process to facilitate feature tuning in generated images. In this sense, the sampling process can be viewed as a downstream task of the self-supervised models.

In comparison to the classifier guidance, which is a popular method for enhancing the performance of the diffusion model, RepG offers multiple advantages. Firstly, our method provides a better representative target than the classifier guidance. The utilization of representative vectors for each class inherently contains valuable information for generative tasks. In contrast, the classifier guidance relies on one-hot vectors representing each class, which offer limited information. This overly compact target leads to reliance on discriminative features within the classifier, which often proves insufficient for generative tasks and raises concerns about potential adversarial effects that could degrade the quality of generated images Dinh et al. (2023b).

Secondly, self-supervised models are trained to generalize well across datasets rather than being tailored to a single task like classifiers. This characteristic helps mitigate the need for noise-aware training of the guidance model, which can be prohibitively expensive, particularly for high-resolution images. Additionally, unlike noise-aware classifiers, self-supervised networks do not require memorizing noise patterns, making the guidance model more lightweight. For instance, our RepG, which leverages ResNet50, achieves efficiency in reducing computational time during sampling.

Thirdly, RepG does not compromise diversity, unlike the classifier guidance approach. While the classifier guidance alters images at the class level to enforce diversity, RepG fine-tunes images at the feature level. Consequently, while the former method encourages the generation of images only with the popular features for each class, RepG preserves most of the image content while modifying faulty features and details.

In summary, our proposed RepG operates distinctively compared to the classifier guidance. As for the classifier-free guidance, while the classifier-free guidance offers a trade-off between quality and diversity, our method focuses on upgrading details or fine-tuning features, as depicted in Figure

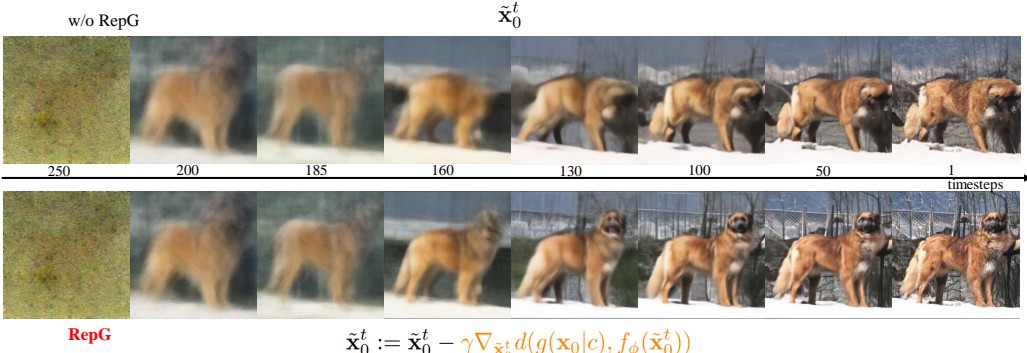

Figure 2: *condition:Leonberg. The top row is the vanilla diffusion sampling process, and the bottom row is the sampling process with our Representative Guidance. From timestep 250 to timestep 185, both of the processes are similar. However, inconsistent features appeared in the vanilla sampling process as the black bubble exists at the head and the tail of the Leonberg at timestep 160. Without RepG, the process struggles to fix inconsistent features for the rest of the process. In contrast, RepG handles the case by removing the inconsistent features and making the image very clear from the time step 130. The RepG sampling process later focuses on improving other details such as hair, background, and surrounding objects. (Dataset: ImageNet256x256/ Diffusion Model: ADM)*

3. Consequently, our RepG can complement the classifier-free guidance to enhance the generation quality further. Combining our method with the classifier-free guidance demonstrates superior performance compared to several SOTA baselines. The contributions of this paper are three-fold:

- Model the incoherence of the diffusion sampling process and introduce a suitable guidance scheme.
- Propose the representative guidance target based on self-supervised pre-trained models.
- Validate the results against a number of state-of-the-art baselines.

## 2 RELATED WORKS

Denoising Diffusion Probabilistic Models (DDPMs) Ho et al. (2020) and their score-based counterparts Song & Ermon (2019); Song et al. (2020b) have become one of the most popular generative models recently and replacing Generative Adversarial Networks (GANs) Odena et al. (2017); Kang et al. (2021); Sauer et al. (2022). The following works Song et al. (2020a); Nichol & Dhariwal (2021); Dhariwal & Nichol (2021); Bao et al. (2022); Lam et al. (2022) improve the models in different perspectives such as time reduction or sampling quality improvement. Recent trends in developing the Diffusion model leveraging the latent space for diffusion and denoising processes such as DiT Peebles & Xie (2023), and Stable Diffusion Rombach et al. (2022) also offer diffusion models with less sampling time with good quality images.

Exposure bias Ning et al. (2023); Yu et al. (2023); Li et al. (2023) is when the noise is accumulated through timesteps due to the lack of ground truth. However, the incoherence problem in this paper has different meanings. Incoherence means a mismatch between two distributions of predicted images at two timesteps that should share the same information. This mismatch results in a gap, allowing incoherent features to be added to the images.

Guidance methods also emerge as essential techniques to boost the performance of generated samples Dhariwal & Nichol (2021); Nichol et al. (2021); Zheng et al. (2022); Dinh et al. (2023a;b); Liu et al. (2023); Bansal et al. (2023). In general, the noise-aware or off-the-shelf classifier/CLIP gradient is utilized to guide the diffusion sampling process to improve its performance in terms of FID. Classifier-free guidance Ho & Salimans (2022) offers a different way to trade off quality with diversity by combining conditional and unconditional diffusion models in the same framework. In Dinh et al. (2023b), the author points out that classifier guidance utilizes the most discriminative features only to do sampling, reducing the generated images' robustness and diversity. However, to achieve superior performance, these methods all give up diversity by significantly modifying details of the image to be close the the most common features of the conditional class. In this manuscript, we propose a guidance method that fixes the details of the image instead of generating another one based on feature-level guidance.

Although ProG Dinh et al. (2023b) solves the problem of diversity suppression by including other classes' features, it still cannot avoid the fact that ProG is still based on discriminative features from a classifier that are not diverse enough for a generative task. Thus, our work utilizes self-supervised models that contain more general information. Self-supervised models Chen et al. (2020b); He et al. (2020); Chen & He (2021); Grill et al. (2020); Chen et al. (2020a) aim to learn representative vectors that contain helpful information about data. While the applications of these models on generative tasks are still limited, this work shows that the pre-trained backbone from a self-supervised model is helpful without any training or fine-tuning. Other self-supervised learning in diffusion models works all aim to fine-tune the diffusion model in a self-supervised manner or utilize the diffusion model as a self-supervised model Hu et al. (2023); Zhang et al. (2024).

## 3 BACKGROUND

**DDPM**: $p_\theta(\mathbf{x}_0) := \int p_\theta(\mathbf{x}_{0:T})d\mathbf{x}_{1:T}$ with $\mathbf{x}_1, \mathbf{x}_2, ..., \mathbf{x}_T$ are latent variables sharing the same dimensionality with the data $\mathbf{x}_0 \sim q(\mathbf{x}_0)$ as the main formulation of DDPMs with $p(\mathbf{x}_T) = \mathcal{N}(\mathbf{x}_T; \mathbf{0}, \mathbf{I})$. The main aim of DDPMs training is to obtain the $p_\theta(\mathbf{x}_{0:T})$ is the *reverse process* following the Markovian property $p_\theta := p(\mathbf{x}_T)\prod_{t=1}^T p_\theta(\mathbf{x}_{t-1}|\mathbf{x}_t)$, where $p_\theta(\mathbf{x}_{t-1}|\mathbf{x}_t) := \mathcal{N}(\mathbf{x}_{t-1}; \mu_\theta(x_t, t), \Sigma_\theta(x_t, t))$. The *reverse process* moves from a total noise image to a clear image. Hence, it is used as a generator in the inference process.

The *forward process* corrupts the original data $\mathbf{x}_0$ to $\mathbf{x}_T$ with Gaussian noise to train the $\theta$ for serving the reverse purpose. This process is a fixed Markov chain $q(\mathbf{x}_{1:T}|\mathbf{x}_0) := \prod_{t=1}^T q(\mathbf{x}_t|\mathbf{x}_{t-1})j$, where $q(\mathbf{x}_t|\mathbf{x}_{t-1}) := \mathcal{N}(\mathbf{x}_t; \sqrt{1-\beta}\mathbf{x}_{t-1}, \beta_t\mathbf{I})$. $\beta_t$ is the fixed variance scheduled from the start of the process.

From the given schedule, distribution of $\mathbf{x}_t$ given $\mathbf{x}_0$ can be derived as:

$$q(\mathbf{x}_t|\mathbf{x}_0) = \mathcal{N}(\mathbf{x}_t; \sqrt{\bar{\alpha}_t}\mathbf{x}_0, (1-\bar{\alpha}_t)\mathbf{I}) \tag{1}$$

Denote $\alpha_t = 1 - \beta_t$ and $\bar{\alpha} = \prod_{s=1}^t \alpha_s$. Reverse from $\mathbf{x}_t$ given $\mathbf{x}_0$, $\mathbf{x}_{t-1}$ distribution is derived as:

$$q(\mathbf{x}_{t-1}|\mathbf{x}_t, \mathbf{x}_0) = \mathcal{N}(\mathbf{x}_{t-1}; \tilde{\boldsymbol{\mu}}_t(\mathbf{x}_t, \mathbf{x}_0), \tilde{\beta}_t\mathbf{I}) \tag{2}$$

Where mean value $\tilde{\boldsymbol{\mu}}_t(\mathbf{x}_t, \mathbf{x}_0) := \frac{\sqrt{\bar{\alpha}_{t-1}}\beta_t}{1-\bar{\alpha}_t}\mathbf{x}_0 + \frac{\sqrt{\alpha_t}(1-\bar{\alpha}_{t-1})}{1-\bar{\alpha}_t}\mathbf{x}_t$ and variance $\tilde{B}_t := \frac{1-\bar{\alpha}_{t-1}}{1-\bar{\alpha}_t}\beta_t$. with reparameterization trick, we can sample the $\mathbf{x}_{t-1}$ as:

$$\mathbf{x}_{t-1} = \frac{(1-\alpha_t)\sqrt{\bar{\alpha}_{t-1}}}{1-\bar{\alpha}_t}\mathbf{x}_0 + \frac{(1-\bar{\alpha}_{t-1})\sqrt{\alpha_t}}{1-\bar{\alpha}_t}\mathbf{x}_t + \sigma_t z \tag{3}$$

Similar to a Variational AutoEncoder Kingma & Welling (2013), the optimization of $\theta$ will be conducted via negative log-likelihood variational bound:

$$\mathbb{E}[-\log p_\theta(\mathbf{x}_0)] \leq \mathbb{E}_q[-\log p(\mathbf{x}_T) - \Sigma_{t \geq 1}\log \frac{p_\theta(\mathbf{x}_{t-1}|\mathbf{x}_t)}{q(\mathbf{x}_t|\mathbf{x}_{t-1})}] \tag{4}$$

We re-write the Eq. 4 as:

$$\mathbb{E}[-\log p_\theta(\mathbf{x}_0)] \leq \mathbb{E}_q[D_{KL}(q(\mathbf{x}_T|\mathbf{x}_0)||p(\mathbf{x}_T)) + \sum_{t>1} D_{KL}(q(\mathbf{x}_{t-1}|\mathbf{x}_t, \mathbf{x}_0)||p_\theta(\mathbf{x}_{t-1}|\mathbf{x}_t)) - \log p_\theta(\mathbf{x}_0|\mathbf{x}_1)]$$

In detail implementation, the $\theta$ is chosen to be parameters of the noise predictor $\epsilon_\theta(\mathbf{x}_t, t)$. The well-trained $\theta$ using Eq. 4 can be used for sampling equation:

$$\mathbf{x}_{t-1} = \frac{1}{\sqrt{\alpha_t}}(\mathbf{x}_t - \frac{1-\alpha_t}{\sqrt{1-\bar{\alpha}_t}}\epsilon_\theta(\mathbf{x}_t, t)) + \sigma_t\mathbf{z} \tag{5}$$

## 4 METHODOLOGY

In this section, we first reformulate the sampling process to analyze the coherence issue. From Eq. 5, we first re-write the formulation of the sampling process as:

$$\mathbf{x}_{t-1} = \frac{(\alpha_t - 1)\sqrt{\bar{\alpha}_{t-1}}}{1-\bar{\alpha}_t}\left(\frac{-\mathbf{x}_t}{\sqrt{\bar{\alpha}_t}} + \frac{\sqrt{1-\bar{\alpha}_t}\epsilon_\theta(\mathbf{x}_t, t)}{\sqrt{\bar{\alpha}_t}}\right) + \frac{(1-\bar{\alpha}_{t-1})\sqrt{\alpha_t}}{1-\bar{\alpha}_t}\mathbf{x}_t + \sigma_t z \tag{6}$$

The complete derivation of Eq.6 can be found in Eq.24 in Appendix. Denote $\tilde{\mathbf{x}}_0^t$ as the prediction of $\mathbf{x}_0$ at time step $t$. From Eq.1, we have $\tilde{\mathbf{x}}_0^t = (\frac{\mathbf{x}_t}{\sqrt{\bar{\alpha}_t}} - \frac{\sqrt{1-\bar{\alpha}_t}\epsilon_\theta(\mathbf{x}_t,t)}{\sqrt{\bar{\alpha}_t}})$ as the prediction of $\tilde{\mathbf{x}}_0$ at the sampling step $t$. This results in a new form of sampling equation:

$$\mathbf{x}_{t-1} = \frac{(1-\alpha_t)\sqrt{\bar{\alpha}_{t-1}}}{1-\bar{\alpha}_t}\tilde{\mathbf{x}}_0^t + \frac{(1-\bar{\alpha}_{t-1})\sqrt{\alpha_t}}{1-\bar{\alpha}_t}\mathbf{x}_t + \sigma_t z \tag{7}$$

The Eq.7 is the sampling from the distribution of $q(\mathbf{x}_{t-1}|\mathbf{x}_t, \tilde{\mathbf{x}}_0)$ with $\tilde{\boldsymbol{\mu}}_t(\mathbf{x}_t, \tilde{\mathbf{x}}_0) = \frac{(1-\alpha_t)\sqrt{\bar{\alpha}_{t-1}}}{1-\bar{\alpha}_t}\tilde{\mathbf{x}}_0^t + \frac{(1-\bar{\alpha}_{t-1})\sqrt{\alpha_t}}{1-\bar{\alpha}_t}\mathbf{x}_t$ and $\sigma_t = \tilde{\beta}$ which is matched with Eq.2 and Eq.3 in the training of the DDPMs.

In the training reverse phase in Eq.3, there is a small assumption that $\mathbf{x}_0 \sim q(\mathbf{x}_0)$. This results in information being passed from timestep $t$ into timestep $t-1$, or the data distribution $q(\mathbf{x}_0)$ is consistent throughout all timesteps. However, this assumption is no longer valid during the sampling step. By assuming $\epsilon_\theta(\mathbf{x}_t, t) \sim \epsilon$, we have:

$$\tilde{\mathbf{x}}_0^t \sim q(\tilde{\mathbf{x}}_0^t|\mathbf{x}_t) = \mathcal{N}(\tilde{\mathbf{x}}_0^t; \frac{-\mathbf{x}_t}{\sqrt{\bar{\alpha}}}, \frac{\sqrt{1-\bar{\alpha}_t}}{\sqrt{\bar{\alpha}_t}}) \tag{8}$$

However, the $q(\tilde{\mathbf{x}}_0^t|\mathbf{x}_t)$ at two different timesteps $t$ are not the same, although they are both used for sampling $\tilde{\mathbf{x}}_0^t$ which is later used for sampling in Eq.7. The illustration of the difference between these distributions can be found in Figure 6 in the Appendix. The assumption that $\mathbf{x}_0 \sim q(\mathbf{x}_0)$ at all timestep is not correct anymore and sample $\mathbf{x}_{t-1}$ from Eq. 7 can not hold. We define the incoherence problem as below:

**Definition 4.1.** *Incoherence* is the mismatch between predicted $\tilde{\mathbf{x}}_0^t$ distributions at different timestep $t$ and mismatch between predicted $\tilde{\mathbf{x}}_0$ distributions with real data distribution $q(\mathbf{x}_0)$.

$$q(\tilde{\mathbf{x}}_0^{t_1}|\mathbf{x}_{t_1}) \neq q(\tilde{\mathbf{x}}_0^{t_2}|\mathbf{x}_{t_2}) \neq q(\mathbf{x}_0) \,\forall t_1, t_2 > 1, t_1 \neq t_2 \tag{9}$$

The incoherence in the sampling process leaves the gap for the inconsistent features resulting from random noise appearing inside the image at some stage of the process. For example, in the top row of Figure 2, we observe the black bubbles at the head and tail of the dog at the $160^{th}$ timestep. The consequence is that the generated samples contain many blur details, inconsistent features, or unnecessary features.

## 4.1 REPRESENTATIVE GUIDANCE

From definition 4.1, gaps of inconsistent features result in poor-quality images. Thus, to solve the incoherence, we need to make the distribution of intermediate samples $q(\tilde{\mathbf{x}}_0^t|\mathbf{x}_t)$ as close as possible to $q(\mathbf{x}_0)$. However, the $q(\mathbf{x}_0)$ is intractable during the sampling process. The intractability would make calculating any distance between these two distributions impossible.

Instead of calculating the direct distance between $q(\tilde{\mathbf{x}}_0^t|\mathbf{x}_t)$ and $q(\mathbf{x}_0)$, we inject features information of $\mathbf{x}_0$ during the sampling process in Eq.7 to force the sampling of $\tilde{\mathbf{x}}_0^t$ to mimic the features of $\mathbf{x}_0$ at every timestep. First, we denote the $f_\phi(\mathbf{x}_0)$ as features extractor, parameterized by $\phi$, for $\mathbf{x}_0$. Our design aims to force $\tilde{\mathbf{x}}_0^t$ to have similar features $f_\phi(\tilde{\mathbf{x}}_0^t)$ as $\mathbf{x}_0$. We denote $d(f_\phi(\mathbf{x}_0), f_\phi(\tilde{\mathbf{x}}_0^t))$ as the distance between two features.

Once again, since $q(\mathbf{x}_0)$ is intractable, $f_\phi(\mathbf{x}_0)$ is also intractable. To address this, instead of representing $f_\phi(\mathbf{x}_0)$ at an instance-wise level, we encode the features of the entire dataset at a class-wise level using $g(\mathbf{x}_0|c)$. Here, $g(\mathbf{x}_0|c)$ is defined as an operation on the set of $f_\phi(\mathbf{x}_0^*) \mid \mathbf{x}_0^* \sim q(\mathbf{x}_0|c)$, where $c \in C$ denotes the class. The feature distance is then transformed into $d(g(\mathbf{x}_0|c), f_\phi(\tilde{\mathbf{x}}_0^t))$.

Given class $c$ and data $\mathbf{x}_0$, $g(\mathbf{x}_0|c)$ results in the set of vectors representing the features of class $c$. Specifically, $g(\mathbf{x}_0|c) = \{r_1^c, r_2^c, \ldots, r_n^c\}$, where $n$ denotes the number of vectors required to represent class $c$. With $C$ classes, the entire dataset is encoded as $V = \{g(\mathbf{x}_0|1), g(\mathbf{x}_0|2), \ldots, g(\mathbf{x}_0|C)\}$. This representation encoding enables the model to store a compact set of representation vectors $g(\mathbf{x}_0|c)$ for each class $c$, rather than storing the representative vectors $f_\phi(\mathbf{x}_0)$ for the entire dataset. The selection of $g$ and $f$ will be discussed in Section 4.2.

As a result, at each time step, given class $c$, we refine the predicted $\tilde{\mathbf{x}}_0$ through the equation below:

$$\tilde{\mathbf{x}}_0^t := \tilde{\mathbf{x}}_0^t - \gamma \nabla_{\tilde{\mathbf{x}}_0^t} d(g(\mathbf{x}_0|c), f_\phi(\tilde{\mathbf{x}}_0^t)), \tag{10}$$

where $\gamma$ is the guidance scale.

From Eq.10 and 7, given class $c$, the new sampling process with coherence is demonstrated in Eq. 11:

$$\mathbf{x}_{t-1} = \frac{(1-\alpha_t)\sqrt{\bar{\alpha}_{t-1}}}{1-\bar{\alpha}_t}\tilde{\mathbf{x}}_0^t + \frac{(1-\bar{\alpha}_{t-1})\sqrt{\alpha_t}}{1-\bar{\alpha}_t}\mathbf{x}_t + \sigma_t z - \frac{(1-\alpha_t)\sqrt{\bar{\alpha}_{t-1}}}{1-\bar{\alpha}_t}\gamma \nabla_{\tilde{\mathbf{x}}_0^t} d(g(\mathbf{x}_0|c), f_\phi(\tilde{\mathbf{x}}_0^t)) \tag{11}$$

with $\tilde{\mathbf{x}}_0^t = (\frac{\mathbf{x}_t}{\sqrt{\bar{\alpha}_t}} - \frac{\sqrt{1-\bar{\alpha}_t}\epsilon_\theta(\mathbf{x}_t, c, t)}{\sqrt{\bar{\alpha}_t}})$. The guidance features from $\mathbf{x}_0$ provide a consistent and reliable target for $\tilde{\mathbf{x}}_0$ to avoid the incoherence problem. We will discuss the similarity between Eq.11 and a Stochastic Gradient Descent process in Appendix D.

The choice of distance $d$ can be varied. The rest of this section 4.1 mainly discusses the options of $d$.

**Negative Cosine similarity:** At each timestep, we sample a vector $r_t^c \sim g(\mathbf{x}_0|c)$. The two vectors $f_\phi(\tilde{\mathbf{x}}_0^t)$ and $r_t^c$ can be matched via a negative cosine similarity loss as below:

$$\mathcal{L}_{\text{cs}}(f_\phi(\tilde{\mathbf{x}}_0^t), r_t^c) = -\frac{f_\phi(\tilde{\mathbf{x}}_0^t) \times r_t^c}{\|f_\phi(\tilde{\mathbf{x}}_0^t)\|\|r_t^c\|} \tag{12}$$

**Contrastive loss:** Apart from negative cosine similarity, the contrastive loss has also been used in many works on representative learning have presented He et al. (2020); Chen et al. (2020b;a). The contrastive loss in our work is more toward the supervised contrastive rather than instance contrastive. We can define a positive pair as two vectors with the same classes and a negative pair as two vectors with different classes. When sampling the image in class $c$, the loss for contrastive matching is:

$$\mathcal{L}_{\text{ct}}(f_\phi(\tilde{\mathbf{x}}_0^t), V) = \frac{\exp\frac{f_\phi(\tilde{\mathbf{x}}_0^t) \times r_t^c}{H}}{\sum_{i=1, i\neq c}^{C} \exp\frac{f_\phi(\tilde{\mathbf{x}}_0^t) \times r_t^i}{H}} \tag{13}$$

where $H$ is the softmax temperature.

Replacing the matching equation in Eq.12 and Eq.13 into Eq.11 as $\mathcal{L}$ , we have the final sampling guidance equation:

$$\mathbf{x}_{t-1} = \frac{(1-\alpha_t)\sqrt{\bar{\alpha}_{t-1}}}{1-\bar{\alpha}_t}\tilde{\mathbf{x}}_0^t + \frac{(1-\bar{\alpha}_{t-1})\sqrt{\alpha_t}}{1-\bar{\alpha}_t}\mathbf{x}_t + \sigma_t z - \frac{(1-\alpha_t)\sqrt{\bar{\alpha}_{t-1}}}{1-\bar{\alpha}_t}\gamma \nabla_{\tilde{\mathbf{x}}_0^t}\mathcal{L}(f_\phi(\tilde{\mathbf{x}}_0^t), V) \tag{14}$$

## 4.2 REPRESENTATIVE TARGETS

In section 4.1, we have discussed a coherent guidance method given representative information from a class $c$. This section will discuss the choice of the mapping function $f_\theta$ and the representative information for each class.

The most straightforward way is to use naive classification for the guidance, where a network such as ResNet He et al. (2016) or a noise-aware classifier Dhariwal & Nichol (2021) is selected to be a classifier. This is the case of the classic classifier guidance. The representative vector $g(\mathbf{x}_0|c) \in \{0, 1\}_C$ reduces to a one-hot vector, with C as the number of classes. However, the use of classification as representative information has many shortages. Firstly, the guidance reveals very little detail about the generated images. Since the classifier only processes the discriminative features, many details that are less discriminative for a class are missed when using the classifier gradient to construct the image Dinh et al. (2023b). Secondly, the motivation for using a classifier to construct images in diffusion models is becoming weaker than the use of the classifier-free guidance. Since a conditional diffusion model already had class-conditioned information, the reason for using additional classification information to improve the performance of a conditioned diffusion model seems to be not strong enough to convince the community. As a result, the research community often opts for classifier-free guidance Rombach et al. (2022); Peebles & Xie (2023). Thirdly, the use of classifier guidance is often associated with the very expensive training cost of noisy classifiers.

Self-supervised models are known to be very good at generalizing data agnostic to augmentation/noise and separating image samples on representative spaces according to features He et al. (2020); Chen & He (2021); Jing et al. (2021). Thus, we choose the self-supervised model to be our guidance model. The self-supervised models are pre-trained and we consider the sampling process as a downstream task of the model. Given a real dataset $\mathbf{x}_0$ and a pre-trained self-supervised model $f_\phi$, an instance $x_i^c \in \mathbf{x}_0$ is the $i^{th}$ instance in class $c$ of the dataset. We have $r_i^c = f_\theta(x_i^c)$. The centre of each class on the representative space has the form $\bar{r}^c = \mathbb{E}\, r_i^c$. We assume that the representative instances that are closer to the mean values represent the most important features of the classes. We represent the whole class $c$ via the representative information $g(\mathbf{x}_0|c)$ as below:

$$g(\mathbf{x}_0|c) = \{r_k^c\}_{\times K}, k \in S_c | \sum_{k \in S_c} \frac{r_k^c \bar{r}^c}{\|r_k^c\|\|\bar{r}^c\|} \to \min \tag{15}$$

Where $S_c$ is the list of indexes of $K$ representative vectors that are selected to be the closest to the class mean representative vector $\bar{r}^c$. We will discuss in section 5 the value of $K$ and different schemes to select representative vectors $g(\mathbf{x}_0|c)$ in addition to the "closest" scheme.

Given Eq.15, we have $V = \{g(\mathbf{x}_0|1), g(\mathbf{x}_0|c), ..., g(\mathbf{x}_0|c)\}$ as a set of representative class vectors representing the whole dataset to enable diffusion sampling with coherence using Eq.14. Before the sampling process starts, $V$ will be calculated in advance and stored as the network parameters for sampling.

## 5 EXPERIMENTAL RESULTS

Experiments are conducted to evaluate on ImageNet Deng et al. (2009) dataset with two resolutions 64x64 and 256x256 with 50000 generated samples. We first verify our claims that our proposed RepG helped to improve the details and fix the faulty information in the images qualitatively in section5.1 and quantitatively in section5.2. After that, we will compare quantitatively with other state-of-the-art methods such as BigGAN Brock et al. (2018), ADM Dhariwal & Nichol (2021), PxP Dinh et al. (2023a), ProG Dinh et al. (2023b), EDS Zheng et al. (2022), IDDPM Nichol & Dhariwal (2021), VAQ-VAE-2 Razavi et al. (2019) and Classifier-free guidance (CLSFree) Ho & Salimans (2022). Three baseline diffusion models are leveraged to evaluate the improvement of the proposed Representative Guidance method are ADM Dhariwal & Nichol (2021), IDDPM Nichol & Dhariwal (2021) and DiT Peebles & Xie (2023).

We denote that *ADM* or *IDDPM* as the ADM or IDDPM diffusion model without guidance. *ADM-G* is denoted for ADM with classifier guidance. *PxP, ProG, EDS* are advanced techniques to improve classifier guidance going after "+" sign. *ADM-CLSFree* and *DiT-CLSFree* are denoted for the application of classifier-free guidance on ADM and DiT respectively. *ADM-CLSFree-G* or *DiT-CLSFree-G* are denoted for applying the combination of classifier-free guidance and classifier guidance on ADM and DiT correspondingly.

### 5.1 INCOHERENT FEATURES ALLEVIATION

As discussed in section 4, we observe incoherent features during the sampling process due to the incoherence of $\tilde{\mathbf{x}}_0^t$ at each timestep. This section shows that RepG successfully alleviates the inconsistent features in the generated images following three categories as in Figure 3. In detail, RepG helps to improve the diffusion sampling process by fixing faulty features, removing unnecessary features, and upgrading details.

### 5.2 QUANTITATIVE IMPROVEMENT

This section compares the performance of our proposed RepG guidance with other state-of-the-art baselines as in Table 1.

Firstly, the use of RepG helps to improve the performance of vanilla baselines such as ADM or IDDPM. Apart from the observation in section 5.1 with qualitative improvement, we see a significant improvement in FID/sFID and precision when applying RepG on ADM or IDDPM. Secondly, given the same ADM diffusion model, *ADM + RepG* has a significantly better Recall value than other guidance methods such as *ADM-G, ADM-CLSFree, ADM-G (+PxP, +ProG, +EDS+ProG)* which

Table 1: *Comparison with state-of-the-art generative baselines on ImageNet64x64 and ImageNet256x26. † denotes the obtained score evaluated from generated images from the published repo. ‡ represents the number taken directly from the paper due to the lack of the source code or generated samples. Other values are reproduced from the published source code. The proposed RepG is shown to achieve better results than other state-of-the-art.*

| Model | FID | sFID | Prec | Rec |
|---|---|---|---|---|
| **ImageNet 64x64** | | | | |
| BigGAN† | 4.06 | 3.96 | 0.79 | 0.48 |
| IDDPM | 2.90 | 3.78 | 0.73 | 0.62 |
| IDDPM + RepG | **2.53** | **3.44** | **0.75** | 0.60 |
| ADM | 2.07 | 4.29 | 0.73 | 0.63 |
| ADM + RepG | **1.69** | **3.42** | **0.75** | 0.62 |
| ADM-G | 2.47 | 4.88 | 0.80 | 0.57 |
| ADM-G + PxP | 1.84 | 3.97 | 0.76 | 0.60 |
| ADM-G + ProG | 1.87 | 4.33 | 0.77 | 0.60 |
| ADM-G + EDS + ProG | 1.77 | 4.25 | 0.77 | 0.61 |
| ADM-CLSFree | 1.89 | 4.45 | 0.77 | 0.60 |
| ADM-CLSFree + ProG | 1.91 | 4.51 | 0.76 | 0.60 |
| ADM-CLSFree + RepG | **1.67** | **3.44** | **0.78** | 0.61 |
| **ImageNet 256x256** | | | | |
| BigGAN† | 7.03 | 7.29 | 0.87 | 0.27 |
| DCTrans‡ | 36.51 | 8.24 | 0.36 | 0.67 |
| VQ-VAE-2‡ | 31.11 | 17.38 | 0.36 | 0.57 |
| IDDPM‡ | 12.26 | 5.42 | 0.70 | 0.62 |
| ADM | 10.94 | 6.02 | 0.69 | 0.63 |
| ADM + RepG | **7.83** | **5.79** | **0.72** | 0.61 |
| ADM-G | 4.58 | 5.23 | 0.81 | 0.52 |
| ADM-G + EDS | 3.96 | 5.00 | 0.82 | 0.52 |
| ADM-G + PxP | 4.00 | 5.19 | 0.81 | 0.53 |
| ADM-G + ProG | 4.53 | 5.08 | 0.85 | 0.49 |
| ADM-G + ProG + EDS | 3.84 | 5.00 | 0.83 | 0.51 |
| ADM-CLSFree | 3.76 | 4.45 | 0.77 | 0.53 |
| ADM-CLSFree-G + ProG | 3.81 | 4.46 | 0.77 | 0.53 |
| ADM-CLSFree + RepG | **3.34** | **4.60** | **0.85** | 0.52 |
| DiT-CLSFree | 2.27 | 4.80 | 0.82 | 0.58 |
| DiT-CLSFree-G + ProG | 2.25 | **4.56** | 0.82 | 0.58 |
| DiT-CLSFree + RepG | **2.17** | 4.59 | 0.80 | **0.60** |

Table 2: *We compare the use of different self-supervised models for our representative guidance.*

| Self-sup Model | FID | sFID | Prec | Rec |
|---|---|---|---|---|
| **ImageNet 64x64** | | | | |
| W/o Guidance | 2.07 | 4.29 | 0.73 | 0.63 |
| MoCo-v2 | **1.69** | **3.42** | 0.75 | 0.62 |
| SimSiam | 1.88 | 3.80 | 0.74 | 0.62 |
| Moco-v3 | 1.81 | 3.93 | **0.76** | 0.62 |

Table 3: *Different K values for our representative guidance with several possible values $K = \{1, 5, 10, 15\}$.*

| | FID | sFID | Prec | Rec |
|---|---|---|---|---|
| **ImageNet64x64** | | | | |
| ADM | 2.07 | 4.29 | 0.73 | 0.63 |
| ADM + RepG (K=1) | 1.77 | 3.44 | 0.75 | 0.60 |
| ADM + RepG (K=5) | 1.69 | 3.42 | 0.75 | 0.62 |
| ADM + RepG (K=10) | 1.73 | 3.43 | 0.75 | 0.62 |
| ADM + RepG (K=15) | 1.82 | 3.45 | 0.75 | 0.62 |

Table 4: *We compare the use of two different representative vector selection schemes.*

| | FID | sFID | Prec | Rec |
|---|---|---|---|---|
| **ImageNet64x64** | | | | |
| ADM | 2.07 | 4.29 | 0.73 | 0.63 |
| ADM + RepG | **1.69** | **3.42** | 0.75 | 0.62 |
| ADM + RepG_Rand | 2.04 | 4.17 | 0.74 | 0.62 |

Table 5: *The use of two matching losses used in sampling as mentioned in section 4 affects the performance of the diffusion sampling process. The result indicates the superiority of both of the losses' performances compared to without guidance. The contrastive achieves slightly better than negative cosine similarity loss.*

| Loss | FID | sFID | Prec | Rec |
|---|---|---|---|---|
| **ImageNet 64x64** | | | | |
| W/o Guidance | 2.07 | 4.29 | 0.73 | 0.63 |
| Contrastive | **1.69** | **3.42** | **0.75** | **0.62** |
| Cosine Similarity | 1.75 | 3.57 | **0.75** | 0.60 |

indicates that *RepG* helps to keep the diversity better than other guidance methods (As highlight in pink column) Finally, The combination of *RepG* and *CLSFree* guidance outperforms other state-of-the-art guidance methods such as PxP Dinh et al. (2023a), ProG Dinh et al. (2023b), EDS Zheng et al. (2022) or CLSFree Ho & Salimans (2022).

*Note:* On ImageNet256x256, the RepG improves baseline ADM significantly but lags behind other guidance methods. This is expected as RepG only improves details and keeps diversity while other methods sacrifice diversity to achieve better quality. On ImageNet64x64, RepG outperforms all other guidance methods due to the information in ImageNet64x64 is less than its 256 counterpart and focuses on foreground objects. Improving objects' features is enough to beat other methods.

Classifier guidance failed to improve the performance of classifier-free guidance significantly (*ADM-CLSFree-G+ProG* and *DiT-CLSFree-G+ProG* in Table 1). This is due to the overlapping trade-off essence of the two methods. These two methods do the same thing: trade-off quality with diversity, which offers less improvement when combined. However, RepG successfully improves classifier-free guidance since RepG does a different task: tune the details of the images.

## 5.3 ABLATION STUDY

Section 5.1 and 5.2 have shown qualitative and quantitative improvement compared to previous state-of-the-art baselines. In the Ablation study, we discuss different choices for our models, such as the choice of self-supervised models, the performance of the proposed methods on different

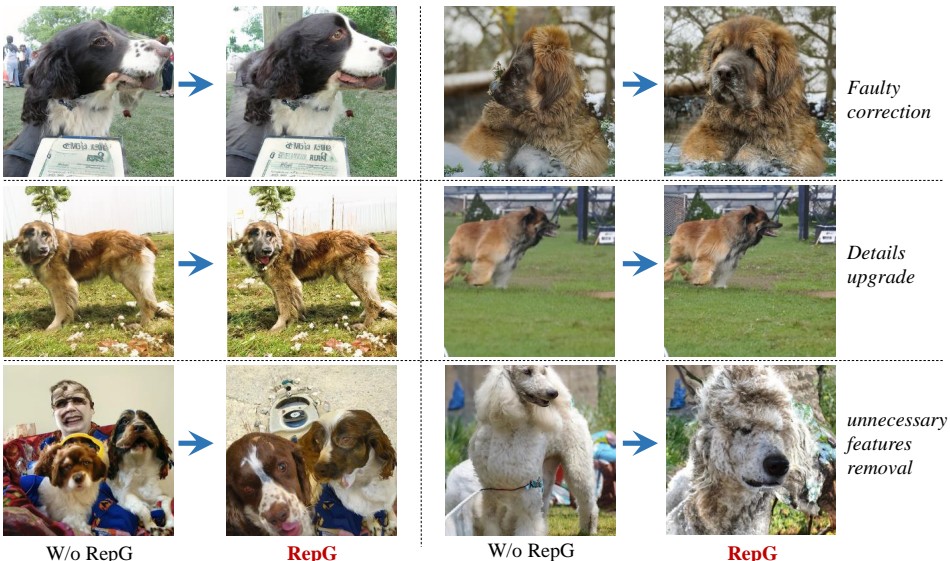

Figure 3: *RepG enhances diffusion sampling in three key ways: (1) correcting faulty features to improve realism (row 1), (2) refining object and background details, such as sharpening the dog's hair and enhancing grass, trees, and fences (row 2), and (3) removing unnecessary elements like a human figure or an incorrect body structure (row 3).* **ImageNet256x256**

guidance scales, the number of representative targets utilized, and the performance comparison between contrastive matching loss (Eq.13) and cosine similarity matching loss (Eq.12).

### 5.3.1 DIFFERENT SELF-SUPERVISED MODELS

In all of the RepG results in Table 1, we use MoCo-v2 Chen et al. (2020b) as the backbone for guidance. This section compares different choices of pretrained self-supervised models in Table 2. In detail, three popular pre-trained self-supervised models are utilized, which are MoCo-v2 Chen et al. (2020b), SimSiam Chen & He (2021), and Moco-v3 Chen et al.. The performance shows that MoCo-v2 achieves the best among the three models. The outperformance of Moco-v2 could be due to the representative information obtained by MoCo-v2 having contrastive information compared to SimSiam, hence obtaining more information about data than just clustering it. Moco-v3 delivers better FID than SimSiam but is still not as good as Moco-v2, yet Moco-v3 offers better Precision.

### 5.3.2 GUIDANCE SCALES EFFECTS

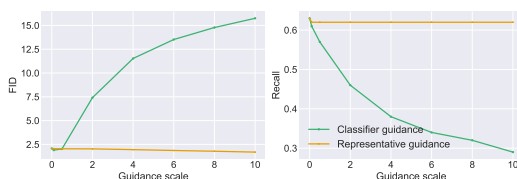

Figure 4: *We compare the Recall trend between the classifier and representative guidance (RepG). RepG shows a much more stable trend in diversity than classifier guidance.*

Similar to the classifier guidance Dhariwal & Nichol (2021); Zheng et al. (2022); Dinh et al. (2023a;b), our RepG can also be controlled by the guidance scale $\gamma$ as in Eq.10. We compare the effect of the guidance scale in the range of $[0, 10]$ with $\gamma = 0.0$ in the diffusion sampling process without any guidance. Figure 4 shows the trend of FID and Recall when increasing the guidance scale. The generation quality of RepG is improved steadily without trading off diversity compared to classifier guidance. Improvement without trading off with diversity is expected since our method mostly keeps the content of the generated images while upgrading the details or fixing faulty features. The effects of increasing the guidance scale can be observed in Figure 5.

### 5.3.3 REPRESENTATIVE TARGETS

This section shows the effect of selecting representative targets for each class.

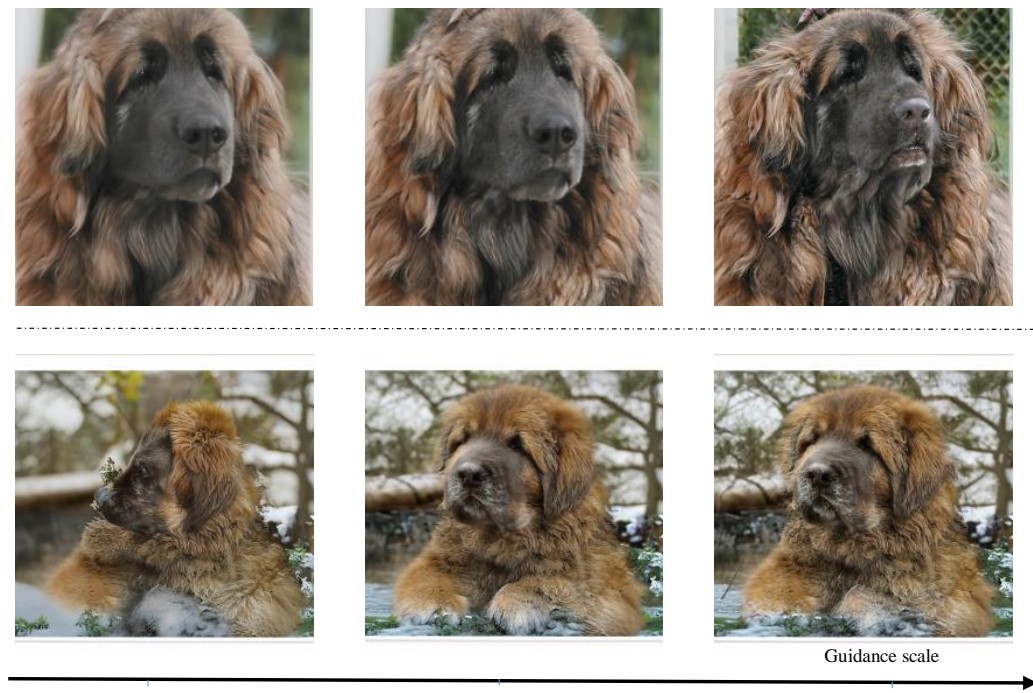

Figure 5: *Ablation study on the RepG guidance scale. Unlike classifier guidance, where increasing the guidance scale shifts the image toward a simpler region (as shown in Figure 7), increasing the RepG guidance scale enhances image details.*

**The $K$ values**: The different choices of $K$ value in Eq.15 affects the performance. The experiment is conducted on ImageNet64x64 as shown in Table 3. As we can see, given K=1, there is only one representative vector for one class, reducing the generated samples' quality and diversity. However, more than five representative vectors per class will confuse the sampling process and downgrade the performance. Understandably, including more representative vectors brings more features to be excluded due to the contrastive loss. The excluded features might include the shared features between classes, which have become common due to the inclusion of more information.

**Selection strategy:** In the previous experiments, representative vectors are selected closest to the mean vector of all vectors belonging to a class. We compare our selection scheme with the random selection scheme in Table 4. RepG_Rand denotes the random selection scheme. From the results, we show that our proposed selection of representative vectors is essential and verify our hypothesis that the vector is close to the mean values of one class bearing crucial features of that class.

### 5.3.4 CONTRASTIVE MATCHING VS COSINE SIMILARITY MATCHING

In section 4.2, we discussed the two losses: the contrastive loss and the cosine similarity loss. Table 5 shows the comparison between the two losses, which show that both of them improve the performance significantly compared to the baseline ADM in Dhariwal & Nichol (2021).

## 6 CONCLUSION

In this work, we formulate the problem of incoherence in the diffusion sampling process, defined as the mismatch between predicted image distribution at two different timesteps. After that, we propose a guidance method named Representative Guidance (RegG). RepG is based on representative information of a class and pre-trained self-supervised models to guide the sampling process. The representative information offers a number of advantages compared to one-hot representation as in classifier guidance, such as rich information and information to avoid incoherence problems.

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

---

**Algorithm 1** DDPM denoising process with representative guidance

---

   **Input:** class labels $y$, classification scale $s$, $V = \{g(\mathbf{x}_0|1), g(\mathbf{x}_0|2), ..., g(\mathbf{x}_0|c)\}$ according to Eq.15
   $\mathbf{x}_T \sim \mathcal{N}(\mathbf{0}, \mathbf{I})$
   pick class $c$ and $g(\mathbf{x}_0|c) \in V$
   **for** $t = T, ..., 1$ **do**
      $z \sim \mathcal{N}(\mathbf{0}, \mathbf{I})$
      $\tilde{\mathbf{x}}_0 \leftarrow \left(\frac{\mathbf{x}_t}{\sqrt{\bar{\alpha}_t}} - \frac{\sqrt{1-\bar{\alpha}_t}\epsilon_\theta(\mathbf{x}_t,t,c)}{\sqrt{\bar{\alpha}_t}}\right)$
      $g \leftarrow \frac{(1-\alpha_t)\sqrt{\bar{\alpha}_{t-1}}}{1-\bar{\alpha}_t}\gamma \nabla_{\tilde{\mathbf{x}}_0^t} \mathcal{L}(f_\phi(\tilde{\mathbf{x}}_0^t), V)$ according to Eq.14
      $\mathbf{x}_{t-1} \leftarrow \frac{1}{\sqrt{\alpha_t}}(\mathbf{x}_t - \frac{1-\alpha_t}{\sqrt{1-\bar{\alpha}_t}}\epsilon_\theta(\mathbf{x}_t,t,c)) + \sigma_t^2 g + \sigma_t z - g$
   **end for**

---

## A   SAMPLING ALGORITHMS

Like DDPMs, our sampling only updates $\tilde{\mathbf{x}}_0^t$ at every time step $t$. We have the set of representative vectors $V$ obtained in advance and stored as the model parameters used for sampling.

The mechanism is the same for latent diffusion, but we will decode the latent vector to $\tilde{\mathbf{x}}_0^t$ first. After that, the process is similar to Algorithm 1.

## B   EXPERIMENTAL DETAILS

All the experiments in this paper are conducted on A100 GPUs 40GB.

We have three hyperparameters in the paper, which are the number of representative vectors $K$ in Eq.15, temperature $H$ in Eq.13 and scale guidance $\gamma$ in Eq.14.

Table 6: *All hyperparameters for producing the results are shown in this table.*

| Model | Datasets | $K$ | $H$ | $\gamma$ |
|---|---|---|---|---|
| Table 1 | | | | |
| IDDPM + RepG | ImageNet64x64 | 5 | 1 | 10.0 |
| ADM + RepG | ImageNet64x64 | 5 | 1 | 10.0 |
| ADM-CLSFree + RepG | ImageNet64x64 | 5 | 1 | 8.0 |
| ADM + RepG | ImageNet256x256 | 10 | 2 | 20.0 |
| ADM-CLSFree + RepG | ImageNet256x256 | 10 | 2 | 20.0 |
| DiT-CLSFree + RepG | ImageNet256x256 | 10 | 2 | 15.0 |
| Table 2 | | | | |
| W/o Guidance | ImageNet64x64 | - | - | 0.0 |
| Moco-v2/SimSiam/Moco-v3 | ImageNet64x64 | 5 | 1 | 10.0 |
| Table 3 | | | | |
| ADM + RepG | ImageNet64x64 | 1,5,10,15 | 1 | 10.0 |
| Table 4 | | | | |
| ADM + RepG /ADM+RepG_RAND | ImageNet64x64 | 5 | 1 | 10.0 |
| Figure 1,2,3,5,6,7 | | | | |
| ADM + RepG | ImageNet256x256 | 10 | 2 | 0.0,20.0, 50.0,80.0 |
| Figure 4 | | | | |
| ADM + RepG | ImageNet64x64 | 5 | 1 | 2.0,4.0, 6.0, 8.0, 10.0 |

Table 7: GPU hours on 1 GPU are needed to generate 50,000 images with 256x256 resolutions. Diffusion Model: ADM/ Datasets: ImageNet256x256

| Model | Computational Cost (GPU hours) |
|---|---|
| No guidance | 171.22 |
| Representative Guidance | 182.36 |
| Classifier Guidance | 247.84 |
| Classifier-free Guidance | 352.89 |

Table 8: GPU hours on 1 GPU are needed to generate 50,000 images with 64x64 resolutions. Diffusion Model: ADM/ Datasets: ImageNet64x64

| Model | Computational Cost (GPU hours) |
|---|---|
| No guidance | 16.71 |
| Representative Guidance | 17.55 |
| Classifier Guidance | 31.52 |
| Classifier-free Guidance | 32.64 |

## C   RUNNING TIME OF REPG COMPARED TO CLASSIFIER GUIDANCE

RepG utilizes a much lighter model compared to noise-aware used in classifier guidance Dhariwal & Nichol (2021). As a result, the calculation of gradients using this model is much lighter compared to the noise-ware classifiers. We have the running time comparison as in Table 7 and 8.

## D   FULL DERIVATION OF EQUATIONS

**Similarity between Eq. 11 and Stochastic Gradient Descent**: We start from Eq.11 as below:

$$\mathbf{x}_{t-1} = \frac{(1-\alpha_t)\sqrt{\bar{\alpha}_{t-1}}}{1-\bar{\alpha}_t}\tilde{\mathbf{x}}_0^t + \frac{(1-\bar{\alpha}_{t-1})\sqrt{\alpha_t}}{1-\bar{\alpha}_t}\mathbf{x}_t + \sigma_t z$$
$$-\frac{(1-\alpha_t)\sqrt{\bar{\alpha}_{t-1}}}{1-\bar{\alpha}_t}\gamma\nabla_{\tilde{\mathbf{x}}_0^t}d(g(\mathbf{x}_0|c), f_\phi(\tilde{\mathbf{x}}_0^t)) \tag{16}$$

with $\tilde{\mathbf{x}}_0^t = (\frac{\mathbf{x}_t}{\sqrt{\bar{\alpha}_t}} - \frac{\sqrt{1-\bar{\alpha}_t}\epsilon_\theta(\mathbf{x}_t,c,t)}{\sqrt{\bar{\alpha}_t}})$. Similarly we have $\mathbf{x}_{t-1} = \sqrt{\bar{\alpha}_{t-1}}\tilde{\mathbf{x}}_0^{t-1} + \sqrt{1-\bar{\alpha}_{t-1}}\epsilon_\theta(\mathbf{x}_{t-1}, c, t-1)$. Thus, we have Eq.16 is equivalent to Eq.17:

$$\tilde{\mathbf{x}}_0^{t-1} = \frac{(1-\alpha_t)}{(1-\bar{\alpha}_t)}\tilde{\mathbf{x}}_0^t - \frac{\sqrt{1-\bar{\alpha}_{t-1}}}{\sqrt{\bar{\alpha}_{t-1}}}\epsilon_\theta(\mathbf{x}_{t-1}, c, t-1) + \frac{(1-\bar{\alpha}_{t-1})\sqrt{\alpha_t}}{(1-\bar{\alpha}_t)\sqrt{\bar{\alpha}_{t-1}}}\mathbf{x}_t + \sigma_t z$$
$$-\frac{(1-\alpha_t)}{1-\bar{\alpha}_t}\gamma\nabla_{\tilde{\mathbf{x}}_0^t}d(g(\mathbf{x}_0|c), f_\phi(\tilde{\mathbf{x}}_0^t))$$
$$=\tilde{\mathbf{x}}_0^t - (\frac{\alpha_t-\bar{\alpha}_t}{1-\bar{\alpha}_t}\tilde{\mathbf{x}}_0^t + \frac{\sqrt{1-\bar{\alpha}_{t-1}}}{\sqrt{\bar{\alpha}_{t-1}}}\epsilon_\theta(\mathbf{x}_{t-1}, c, t-1) - \frac{(1-\bar{\alpha}_{t-1})\sqrt{\alpha_t}}{(1-\bar{\alpha}_t)\sqrt{\bar{\alpha}_{t-1}}}\mathbf{x}_t - \sigma_t z)$$
$$-\frac{(1-\alpha_t)}{1-\bar{\alpha}_t}\gamma\nabla_{\tilde{\mathbf{x}}_0^t}d(g(\mathbf{x}_0|c), f_\phi(\tilde{\mathbf{x}}_0^t)) \tag{17}$$

with $\mathbf{x}_{t-1}$ is obtained from Eq.16.

The Eq.17 has a very close form with a Stochastic Gradient Descent optimization with $\tilde{\mathbf{x}}_0^t$ as parameters and two gradients $\nabla_1 = \frac{\alpha_t-\bar{\alpha}_t}{1-\bar{\alpha}_t}\tilde{\mathbf{x}}_0^t + \frac{\sqrt{1-\bar{\alpha}_{t-1}}}{\sqrt{\bar{\alpha}_{t-1}}}\epsilon_\theta(\mathbf{x}_{t-1}, c, t-1) - \frac{(1-\bar{\alpha}_{t-1})\sqrt{\alpha_t}}{(1-\bar{\alpha}_t)\sqrt{\bar{\alpha}_{t-1}}}\mathbf{x}_t - \sigma_t z$ and $\nabla_2 = \frac{(1-\alpha_t)}{1-\bar{\alpha}_t}\gamma\nabla_{\tilde{\mathbf{x}}_0^t}d(g(\mathbf{x}_0|c), f_\phi(\tilde{\mathbf{x}}_0^t))$. We will show that this Eq.17 has a consistent objective function. From Eq.1 and two timesteps $t_1 < t_2$, we have:

$$\mathbf{x}_{t_1} = \sqrt{\bar{\alpha}_{t_1}}\mathbf{x}_0 + \sqrt{1-\bar{\alpha}_{t_1}}\epsilon_1 \tag{18}$$
$$\mathbf{x}_{t_2} = \sqrt{\bar{\alpha}_{t_2}}\mathbf{x}_0 + \sqrt{1-\bar{\alpha}_{t_2}}\epsilon_2 \tag{19}$$

From $\mathbf{x}_{t_1}$ and $\mathbf{x}_{t_2}$, we have the prediction of $\mathbf{x}_0$ at $t_1$ is $\mathbf{x}_0^{(t_1)}$ and $t_2$ is $\mathbf{x}_0^{(t_2)}$. We have:

$$\tilde{\mathbf{x}}_0^{(t_1)} = \frac{\mathbf{x}_{t_1} - \sqrt{1 - \bar{\alpha}_{t_1}}\epsilon_\theta(\mathbf{x}_{t_1}, t_1)}{\sqrt{\bar{\alpha}_{t_1}}} \tag{20}$$

$$\tilde{\mathbf{x}}_0^{(t_2)} = \frac{\mathbf{x}_{t_2} - \sqrt{1 - \bar{\alpha}_{t_2}}\epsilon_\theta(\mathbf{x}_{t_2}, t_2)}{\sqrt{\bar{\alpha}_{t_2}}} \tag{21}$$

Replace Eq.18 and 19 into Eq.20 and 21, we have:

$$\tilde{\mathbf{x}}_0^{(t_1)} = \mathbf{x}_0 + \frac{\sqrt{1 - \bar{\alpha}_{t_1}}(\epsilon_1 - \epsilon_\theta(\mathbf{x}_{t_1}, t_1))}{\sqrt{\bar{\alpha}_{t_1}}} \tag{22}$$

$$\tilde{\mathbf{x}}_0^{(t_2)} = \mathbf{x}_0 + \frac{\sqrt{1 - \bar{\alpha}_{t_2}}(\epsilon_2 - \epsilon_\theta(\mathbf{x}_{t_2}, t_2))}{\sqrt{\bar{\alpha}_{t_2}}} \tag{23}$$

From Eq.22 and 23, we have at any timestep $t$, the distance between $\tilde{\mathbf{x}}_0^t - \mathbf{x}_0 = \frac{\sqrt{1-\bar{\alpha}_{t_1}}(\epsilon - \epsilon_\theta(\mathbf{x}_t, t))}{\sqrt{\bar{\alpha}_t}}$ which means $||\tilde{\mathbf{x}}_0^t - \mathbf{x}_0|| = \frac{\sqrt{1-\bar{\alpha}_{t_1}}||\epsilon - \epsilon_\theta(\mathbf{x}_t, t)||}{\sqrt{\bar{\alpha}_t}}$. Assuming that $\epsilon_\theta$ is trained to converge, we assume $||\epsilon_\theta(\mathbf{x}_{t_1}, t_1) - \epsilon_1|| \leq ||\epsilon_\theta(\mathbf{x}_{t_2}, t_2) - \epsilon_2||$, because when image is clearer, we also expect the error should be smaller. The extreme case is $||\epsilon_1 - \epsilon_\theta(\mathbf{x}_{t_1}, t_1)|| \approx ||\epsilon_2 - \epsilon_\theta(\mathbf{x}_{t_2}, t_2)|| \approx \Delta$. As a result $||\tilde{\mathbf{x}}_0^{t_1} - \mathbf{x}_0|| = \frac{\sqrt{1-\bar{\alpha}_{t_1}}\Delta}{\sqrt{\bar{\alpha}_{t_1}}}$ and $||\tilde{\mathbf{x}}_0^{t_2} - \mathbf{x}_0|| = \frac{\sqrt{1-\bar{\alpha}_{t_2}}\Delta}{\sqrt{\bar{\alpha}_{t_2}}}$. Since $t_1 < t_2$, $\frac{\sqrt{1-\bar{\alpha}_{t_1}}}{\sqrt{\bar{\alpha}_{t_1}}} < \frac{\sqrt{1-\bar{\alpha}_{t_2}}}{\sqrt{\bar{\alpha}_{t_2}}}$ which means $||\tilde{\mathbf{x}}_0^{(t_1)} - \mathbf{x}_0|| < ||\tilde{\mathbf{x}}_0^{(t_2)} - \mathbf{x}_0|| \forall t_1 < t_2$. Which means that from $T$ to $0$, the sampling process will update $\tilde{\mathbf{x}}_0^t$ so that $||\tilde{\mathbf{x}}_0^t - \mathbf{x}_0|| \to \min$. We have the first gradient of the Eq.17 is $\nabla_1 = \frac{\alpha_t - \bar{\alpha}_t}{1 - \bar{\alpha}_t}\tilde{\mathbf{x}}_0^t + \frac{\sqrt{1-\bar{\alpha}_{t-1}}}{\sqrt{\bar{\alpha}_{t-1}}}\epsilon_\theta(\mathbf{x}_{t-1}, c, t-1) - \frac{(1-\bar{\alpha}_{t-1})\sqrt{\alpha_t}}{(1-\bar{\alpha}_t)\sqrt{\bar{\alpha}_{t-1}}}\mathbf{x}_t - \sigma_t z$.

We can easily see the second gradient is the $\nabla_2 = \frac{(1-\alpha_t)}{1-\bar{\alpha}_t}\gamma\nabla_{\tilde{\mathbf{x}}_0^t}d(g(\mathbf{x}_0|c), f_\phi(\tilde{\mathbf{x}}_0^t))$ to minimize the distance $d(g(\mathbf{x}_0|c), f_\phi(\tilde{\mathbf{x}}_0^t))$.

Thus, we can conclude that the sampling process as Eq.11 is a process of Stochastic Gradient Descent to optimize two objectives. The first objective is $\min_{\tilde{\mathbf{x}}_0^t}||\tilde{\mathbf{x}}_0^t - \mathbf{x}_0||$ and the second objective is $\min_{\tilde{\mathbf{x}}_0^t}d(g(\mathbf{x}_0|c), f_\phi(\tilde{\mathbf{x}}_0^t))$.

**Full derivation of Eq.6:** Eq.6 can be fully derived as below:

$$\begin{aligned}
\mathbf{x}_{t-1} &= \frac{1}{\sqrt{\alpha_t}}\mathbf{x}_t - \frac{1-\alpha_t}{\sqrt{1-\bar{\alpha}_t}}\epsilon_\theta(\mathbf{x}_t, t) + \sigma_t z \\
&= \left(\frac{1-\alpha_t}{(1-\bar{\alpha})\sqrt{\alpha_t}}\mathbf{x}_t + \frac{(1-\bar{\alpha}_{t-1})\sqrt{\alpha_t}}{1-\bar{\alpha}_t}\mathbf{x}_t\right) - \frac{1-\alpha_t}{\sqrt{1-\bar{\alpha}_t}}\epsilon_\theta(\mathbf{x}_t, t) + \sigma_t z \\
&= \frac{1-\alpha_t}{1-\bar{\alpha}_t}\left(\frac{\mathbf{x}_t}{\sqrt{\alpha_t}} - \frac{\sqrt{1-\bar{\alpha}_t}}{\sqrt{\alpha_t}}\epsilon_\theta(\mathbf{x}_t, t)\right) + \frac{(1-\bar{\alpha}_{t-1})\sqrt{\alpha_t}}{1-\bar{\alpha}_t}\mathbf{x}_t + \sigma_t z \\
&= \frac{(1-\alpha_t)\sqrt{\bar{\alpha}_{t-1}}}{1-\bar{\alpha}_t}\left(\frac{\mathbf{x}_t}{\sqrt{\bar{\alpha}_t}} - \frac{\sqrt{1-\bar{\alpha}_t}\epsilon_\theta(\mathbf{x}_t, t)}{\sqrt{\bar{\alpha}_t}}\right) + \frac{(1-\bar{\alpha}_{t-1})\sqrt{\alpha_t}}{1-\bar{\alpha}_t}\mathbf{x}_t + \sigma_t z \tag{24}
\end{aligned}$$

## E  $\tilde{\mathbf{x}}_0$ DISTRIBUTION

Figure 6 shows the difference in the distributions of $\tilde{\mathbf{x}}_0^t$ at different timesteps.

## F  CLASSIFER GUIDANCE DIVERSITY SUPPRESSION

Similar to Dinh et al. (2023b), we reproduce the diversity suppression of classifier guidance as in Figure 8, 9 and 10.

## G  MORE QUALITATIVE RESULTS COMPARISON FOR REPG

Figure 11, 12, 13 and 14 shows more examples of how RepG can help to fix details in the generated images

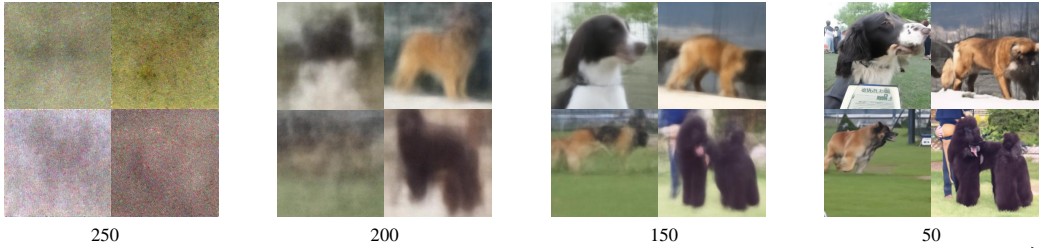

Figure 6: *Visualization of $\tilde{\mathbf{x}}_0^t$ at different timesteps. $\tilde{\mathbf{x}}_0^t$ has different distributions when $t$ varies. The earlier timesteps have less information, while the later stage has clearer views of the images.*

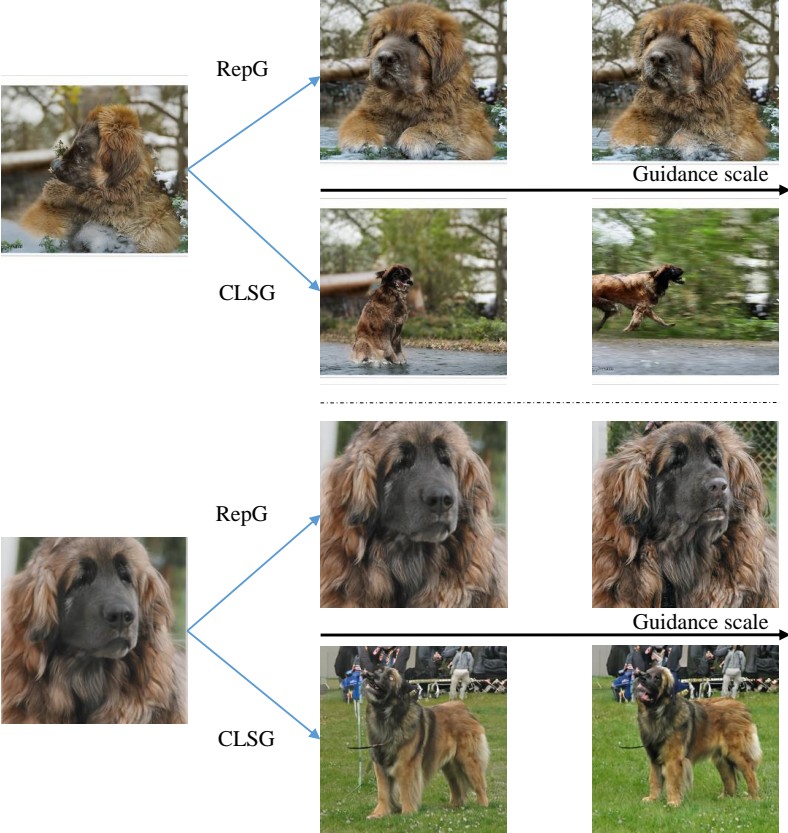

Figure 7: *RepG edit details in the image while Classifier Guidance (CLSG) generates another image with a good discriminative feature. However, sometimes, the over-exploiting of discriminative features results in the lack of robustness features in the output.*

## H MORE SAMPLES WITH REPG

Figure 15 shows several samples by DiT combined with RepG.

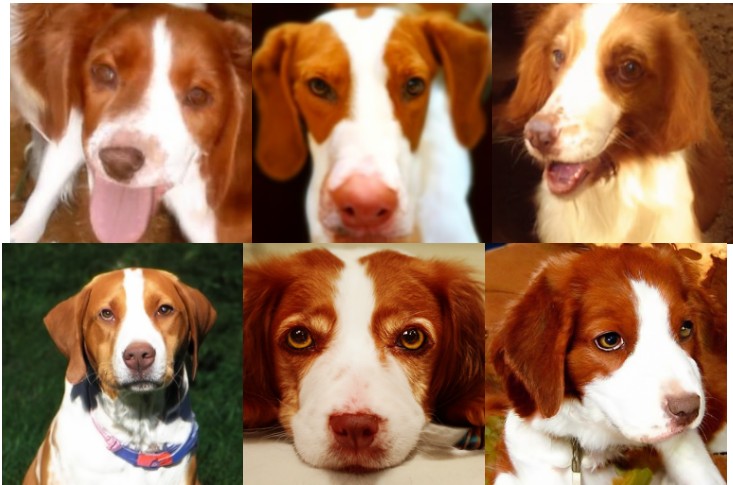

Figure 8: *Classifier guidance utilizes the same style to repeat features to all generated images. This is due to the overexploitation of discriminative features (front-face features) reducing the diversity of the diffusion model*

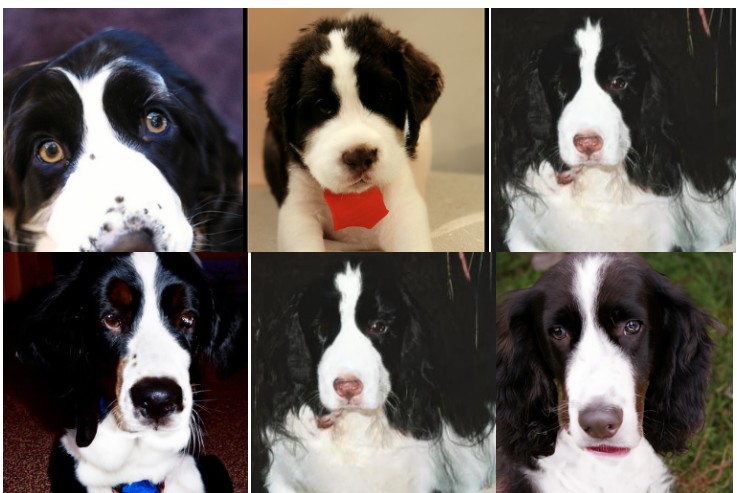

Figure 9: *Classifier guidance utilizes the same style to repeat features to all generated images. This is due to the overexploitation of discriminative features (front-face features) reducing the diversity of the diffusion model*

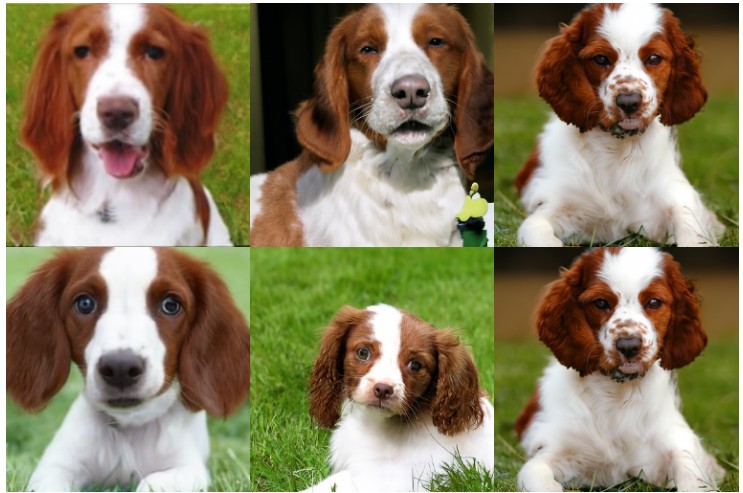

Figure 10: *Classifier guidance utilizes the same style to repeat features to all generated images. This is due to the overexploitation of discriminative features (lie-in-bed features) reducing the diversity of the diffusion model*

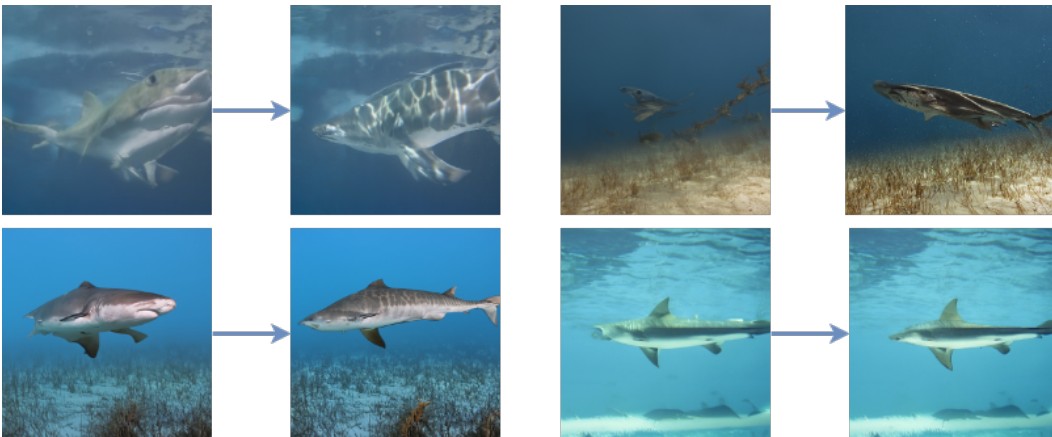

Figure 11: *ImageNet256x256/class: tiger shark. The images on the left, shown before the arrow, are the erroneous outputs generated by ADM, while the images on the right, after the arrow, depict the corrections made using RepG. The examples show that RepG can improve the details and fix the erroneous features.*

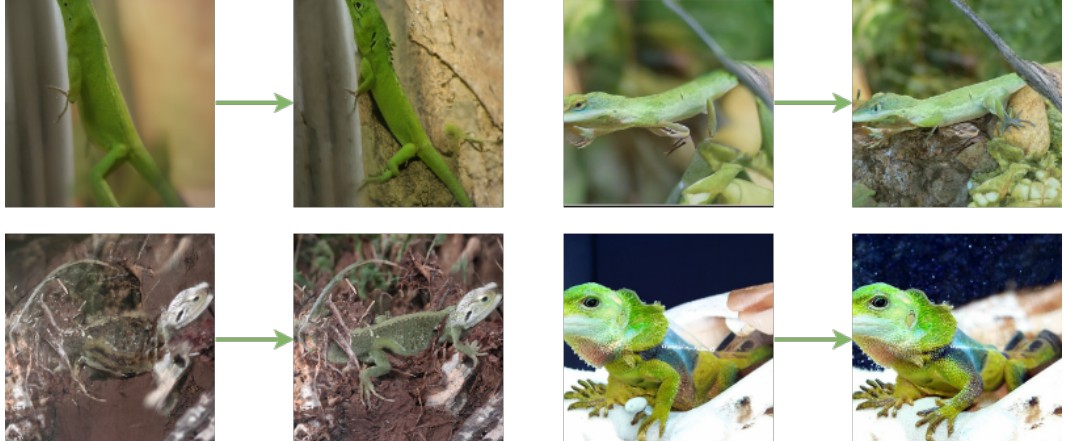

Figure 12: *ImageNet256x256/class: green lizard. The images on the left, shown before the arrow, are the erroneous outputs generated by ADM, while the images on the right, after the arrow, depict the corrections made using RepG. The examples show that RepG can improve details/background, remove unnecessary features and fix erroneous features.*

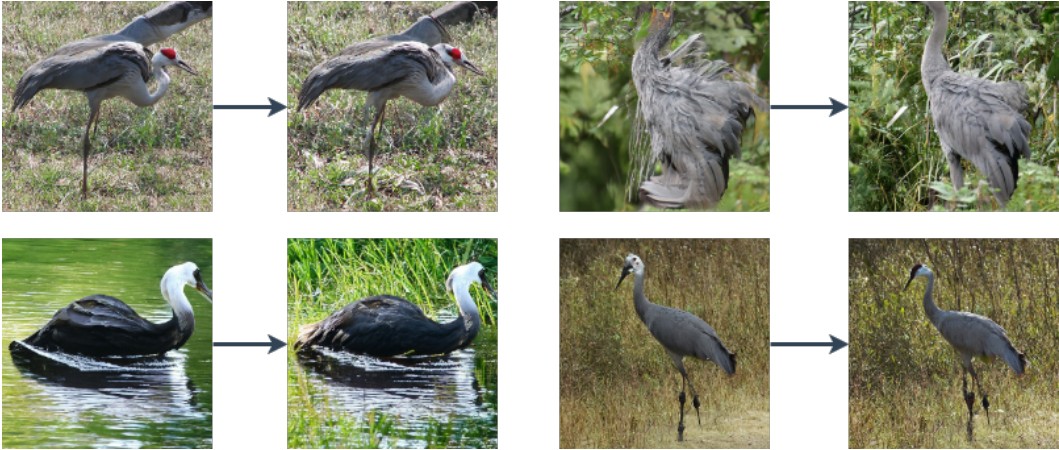

Figure 13: *ImageNet256x256/class: crane. The images on the left, shown before the arrow, are the erroneous outputs generated by ADM, while the images on the right, after the arrow, depict the corrections made using RepG.The examples show that RepG can upgrade details, modify faulty features.*

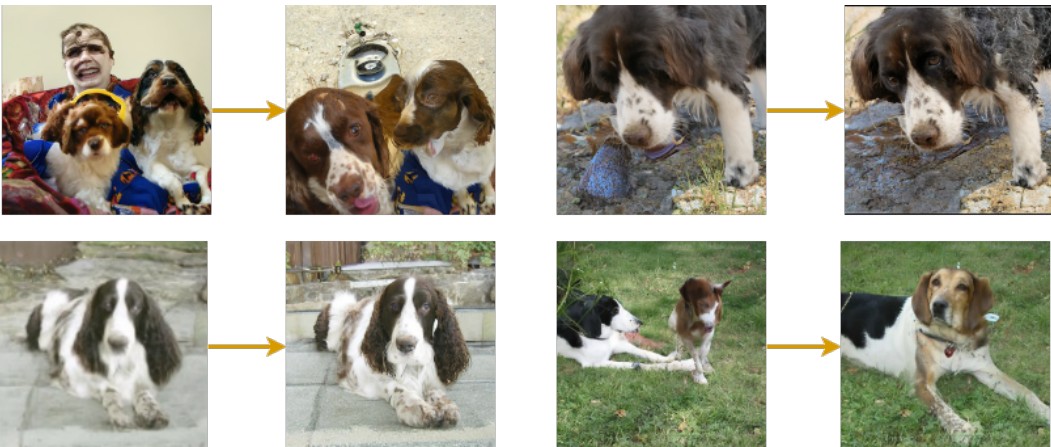

Figure 14: *ImageNet256x256/class: English springer. The images on the left, shown before the arrow, are the erroneous outputs generated by ADM, while the images on the right, after the arrow, depict the corrections made using RepG. The examples show that RepG can improve details/background, remove unnecessary features and fix erroneous features.*

**Goldfish**

**Crampfish**

**Jellyfish**

**Snail**

**Red fox**

**Polar bear**

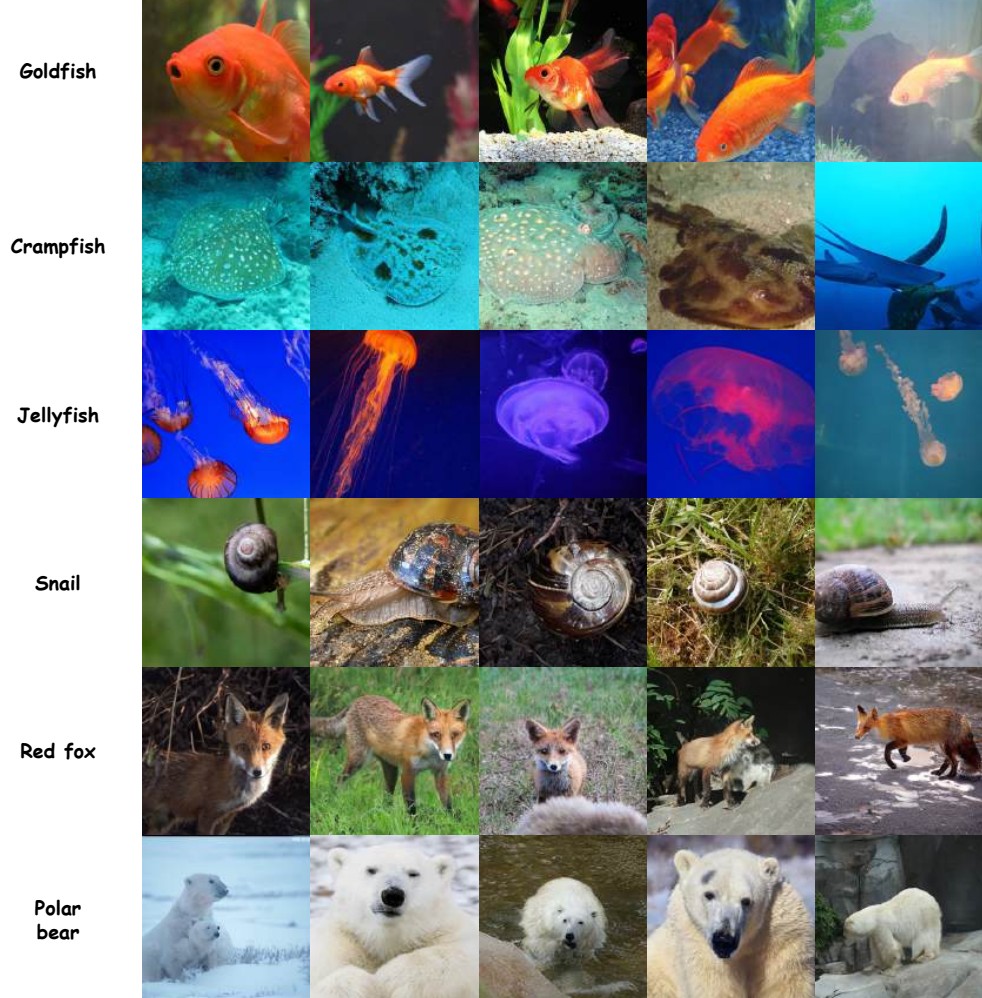

Figure 15: *Sampling by DiT with RepG for several classes.*

