# OpenReview forum: "Representative Guidance: Diffusion Model Sampling with Coherence"
_ICLR.cc/2025/Conference — ICLR 2025 Spotlight_

### Official Review · Reviewer_9ZFf · 2024-11-01

**Soundness:** 4
**Presentation:** 4
**Contribution:** 4
**Rating:** 8
**Confidence:** 4

**Summary:**

The paper proposes a solution for improving diffusion sampling process that involves tuning image features at each time step to rectify incoherent features. For this purpose, the authors introduce a guidance scheme called Representative Guidance (RepG), which leverages information from representative vectors to steer the sampling process towards a coherent direction. Moreover, RepG represents each class through a set of representative vectors containing features specific to that class. To control the optimal representative information, RepG
employs self-supervision as guidance model. The gradients derived using the pre-trained self-supervised model are directly integrated into the sampling process to facilitate feature tuning in the generated images. Extensive experimental results demonstrate that RepG achieves superior performance in image generation.

**Strengths:**

- The paper is clearly written and easy to follow
- The related work covers most of the relevant papers in the field
- Experimental validation is extensive and demonstrates the superiority of the proposed approach

**Weaknesses:**

- Some details require more clarification

**Questions:**

Here are my concerns:
- Please provide more details on how the representative vectors in RepG capture more valuable information for generative tasks than the one-hot vectors used in classifier guidance.
- You stated in the paper that RepG modifies "faulty" features in the generated images. How does your appoach identify which features are faulty? It could be possible that some key features could be mistakenly modified, thus affecting the image quality?
- You mentioned in the paper that RepG is computationally efficient. Could you provide comparative metrics on the computational savings over traditional classifier-guided methods (including for instance image generation at different resolutions)?
- Definition 4.1: t1 and t2 should be consecutive time steps?
- Equation 13: In the denominator, the superscript of 'r' should be 'i', not 'c'.

---

> ### Author Response · Authors · 2024-11-23
> **Thank you for your comments**
>
> We thank the reviewer for the thoughtful comments.
>
> We answer your main concerns below:
>
> ### Q1. More details on how representative vectors in RepG capture more valuable information for generative tasks than one-hot vector used in classifier guidance.
>
>
> A one-hot vector by itself does not carry much meaningful information. Instead, it serves as a signal for the model to extract discriminative features stored in the classifier. However, this approach introduces several challenges:
>
> 1. **Variability in Representation**: Within a single class, not all images result in a near one-hot representation after passing through a classifier. Forcing images to conform to a one-hot representation can lead to significant and often unnecessary changes in the images, solely to satisfy this constraint.
>
> 2. **Limited Utilization of Features**: Discriminative features stored in the classifier are primarily optimized for classification tasks. These features typically represent only the most common characteristics of objects, whereas generative features encompass richer details, including nuances in both the foreground and background.
>
> **Representative vectors**, on the other hand, offer several advantages:
>
> 1. **Flexible Representation**: Representative vectors enable more diverse and nuanced generated samples, avoiding the many samples with the same target phenomenon like one-hot vectors.
>
> 2. **Enhanced Generative Features**: Unlike one-hot vectors, representative vectors themselves contain meaningful information, which contributes to improved generative feature quality.
>
> 3. **Feature-Level Adjustments**: Using representative vectors allows modifications at the feature level rather than the class level. This leads to finer detail enhancements without shifting the image into the 'easy' regions guided by classifier-based adjustments.
>
>
> ### Q2. RepG modifies "faulty" features in the generated images. How does your approach identify which features are faulty? It could be possible that some key features could be mistakenly modified, thus affecting the image quality.
>
> In most of the results presented in this paper, we primarily use the contrastive loss (Eq. 13). The contrastive loss serves two key purposes in the construction of the sample $x^{t}_0$:
>
> 1. Staying closer to positive features.
> 2. Pushing negative features farther away.
>
> We define **positive features** as those belonging to a specific class, while **negative features** are those that do not belong to that class. These features are represented by representative vectors. Consequently, any feature that is not sufficiently close to the representative features of a class is pushed away, effectively eliminating "faulty" features.
>
> ### Q3. Computational Efficiency
>
> Please find the computational efficiency below tables (R2 and R3) (We use Nvidia 4090RTX for this experiment):
>
> | Model                    | Computational cost (GPU hours) |
> |--------------------------|--------------------------------|
> | No guidance              | *171.22*                         |
> | Representative guidance  | **182.36**                     |
> | Classifier guidance      | 247.84                         |
> | Classifier-free guidance | 352.89                         |
>
> Table R2. GPU hours on 1 GPU are needed to generate 50,000 images with 256x256 resolutions. Diffusion Model: ADM/ Datasets: ImageNet256x256
>
> | Model                    | Computational cost (GPU hours) |
> |--------------------------|--------------------------------|
> | No guidance              | *16.71*                        |
> | Representative guidance  | **17.55**                      |
> | Classifier guidance      | 31.52                          |
> | Classifier-free guidance | 32.64                          |
>
> Table R3. GPU hours on 1 GPU needed for generating 50000 images with 64x64 resolutions. Diffusion Model: ADM/ Datasets: ImageNet64x64
>
> The primary computational expense of classifier guidance arises from the backpropagation through the noise-aware classifier. In contrast, representative guidance relies on much lighter models, such as ResNet-50, significantly reducing the sampling time.
>
> ### Q4. Definition 4.1: t1 and t2 should be consecutive time steps?
>
> At any two timesteps $t_1$ and $t_2$, the distribution of the predicted $\tilde{x}^{t_1}_0$ and $\tilde{x}^{t_2}_0$ do not align. So Definition 4.1 holds for any $t_1$ and $t_2$.
>
> ### Q5. Equation 13: In the denominator, the superscript of 'r' should be 'i', not 'c'
>
> We have updated the equation in the paper as below:
>
> $\mathcal{L}_{\text{ct}}(f _{\phi}(\tilde{x}_0^t), V) = \frac{\exp{\frac{f _{\phi}(\tilde{x}_0^t) \times  r^c_t}{H}}}{\sum^{c} _{i=1, I \neq c} \exp{\frac{f _{\phi}(\tilde{x} _0^t) \times  r^i _t}{H}}} $

---

> ### Author Response · Authors · 2024-11-25
> **Thank you a lot for your comments**
>
> Dear Reviewer 9ZFf,
>
> Thank you for your thoughtful comments and valuable feedback. We have carefully addressed all your concerns in our revisions.
>
> May we kindly ask if you have any further questions or feedback regarding the paper? We would be happy to provide additional clarifications if needed.
>
> Thank you once again for your time and consideration.

---

> > ### Comment · Reviewer_9ZFf · 2024-11-26
> > **Official Comment by Reviewer 9ZFf**
> >
> > The authors satisfactorily addressed my concerns, therefore I maintain my initial rating.

---

> > > ### Author Response · Authors · 2024-11-27
> > > **Thank you for your support of our work**
> > >
> > > Dear Reviewer 9ZFf,
> > >
> > > Thank you very much for your thoughtful comments and positive feedback. Your insights are greatly appreciated and have been extremely helpful in improving the manuscript.
> > >
> > > Best regards,

---

### Official Review · Reviewer_eBdC · 2024-11-03

**Soundness:** 3
**Presentation:** 2
**Contribution:** 2
**Rating:** 6
**Confidence:** 3

**Summary:**

The authors proposed an approach, namely representative guidance (RepG) to make the sampling process of diffusion model along a consistent direction. The proposed RepG is a plugin module, which can be easily adapted to existing framework. Particularly, RepG adopts the self-supervised representation to achieve the aim of consistent sampling direction of diffusion. The proposed approach is evaluated on publicly available ImageNet dataset, which presents improvements to existing diffusion models.

**Strengths:**

1. The paper is easy to follow.
2. The approach has been evaluated on publicly available dataset, which makes the experimental results convincing.

**Weaknesses:**

1. There are lots of typos in the paper, such as the caption of Table 1: ImageNet 256x26 -> ImageNet 256x256. The authors should carefully proof-read the manuscript.
2. Some important baselines are missing from Table 1, such as the comparison between RepG and the existing approaches with IDDPM. The comparison with ADM backbone seems to be more comprehensive. The authors should illustrate the reason for such an experimental setting.
3. With ADM backbone, the experimental results are also difficult to directly compare. Too much different setting of the ADM variant. The authors may consider to simply compare the methods with ADM + X (where X is RepG / PxP and the others).
4. Why the performance of MoCo-v3 is lower than MoCo-v2 in Table 2?
5. Please include the visualization results generated by existing approaches as well.

**Questions:**

Please refer to weakness section for details.

---

> ### Author Response · Authors · 2024-11-23
> **We have addressed all of your concerns**
>
> Dear Reviewer eBdC,
>
> We have addressed all of your concerns about our writing
>
> ## Q1. Typos in the paper
>
> We have fixed the caption of Table 1 and checked with other typos including:
>
> * L181: close round bracket in the equation
> * L132: Denosing -> Denoising
> * L155: diversity supersession -> diversity suppression
> * L160: This -> this
> * L254: a an -> an
> * Eq13: c -> i
>
>
> ## Q2. Lack of comparison between RepG and the existing approaches with IDDPM. The comparison with ADM backbone seems to be more comprehensive. The authors should illustrate the reason for such an experimental setting.
>
> **Lack of comparison of RepG on IDDPM**: We have evaluated the performance of RepG on IDDPM using ImageNet64x64. However, we could not provide results for ImageNet256x256 on IDDPM because the IDDPM paper [1] does not include a pretrained diffusion model for this resolution. Most of their baselines of IDDPM are on 64x64 and 128x128. Without access to the IDDPM checkpoint, we were unable to perform sampling for both the baseline and guidance, which explains the absence of these comparisons on ImageNet256x256.
>
>
> **The comparison with ADM backbone seems to be more comprehensive**: This is because:
> 1. The ADM paper [2] was the first to introduce classifier guidance.
> 2. Subsequent variants of classifier guidance [3][4][5] have all been built upon the ADM diffusion backbone. As a result, most comparisons for classifier guidance variants are based on ADM.
> 3. Additionally, for pixel-space diffusion models, ADM achieves state-of-the-art results, significantly outperforming other models, making it a more robust and comprehensive benchmark for evaluation.
>
>
>
>
> ## Q3. With ADM backbone, the experimental results are also difficult to directly compare. Too much different setting of the ADM variant. The authors may consider to simply compare the methods with ADM + X
>
> Thank you for your suggestions. We have updated Table 1 in the revision, with each resolution having two separate experiments. One is above the bold horizontal line and one is below the **bold** horizontal line.
>
> The above **bold** horizontal line shows the improvements of RepG when combines with ADM/IDDPM diffusion models. The below **bold** horizontal line shows the SOTA achieved by using RepG.
>
> We can separate into two tables in the camera ready version since we will have one more page. The current limitation in the page number does not allow us to separate into two tables.
>
> ## Q4. Why the performance of MoCo-v3 is lower than MoCo-v2 in Table 2?
>
> MoCo v3 does not necessarily outperform MoCo v2. According to the MoCo v3 paper, its primary aim is to enhance stability when used with Vision Transformer (ViT) backbones, rather than to achieve superior representative quality[5].
>
> Both MoCo v3 and SimSiam eliminate the use of a negative queue during training. While this simplification aids in achieving linear separation of instance representations, it reduces the model's ability to capture information about other instances in the feature space. In contrast, MoCo v2 utilizes contrastive loss to explicitly push apart "negative features," enabling a richer and more discriminative feature space. The absence of this mechanism in MoCo v3 and SimSiam makes them less effective in generative tasks and in our guidance method, where contrastive loss plays a critical role in supporting the generation of high-quality features.
>
> [1] Nichol, Alexander Quinn, and Prafulla Dhariwal. "Improved denoising diffusion probabilistic models." International conference on machine learning. PMLR, 2021.
>
> [2] Dhariwal, Prafulla, and Alexander Nichol. "Diffusion models beat gans on image synthesis." Advances in neural information processing systems 34 (2021): 8780-8794.
>
> [3] Dinh, Anh-Dung, Daochang Liu, and Chang Xu. "PixelAsParam: A gradient view on diffusion sampling with guidance." International Conference on Machine Learning. PMLR, 2023.
>
> [4] Zheng, Guangcong, et al. "Entropy-driven sampling and training scheme for conditional diffusion generation." European Conference on Computer Vision. Cham: Springer Nature Switzerland, 2022.
>
> [5] Chen, Xinlei, Saining Xie, and Kaiming He. "An empirical study of training self-supervised vision transformers." Proceedings of the IEEE/CVF international conference on computer vision. 2021.
>
> ## Q5. Please include the visualization results generated by existing approaches as well.
>
> As already shown in Figure 7, the existing approach fails to enhance image features effectively. Instead, it seeks a more comfortable zone, altering the entire image and reducing diversity. To provide further evidence of this issue, we now include Figures 8, 9, and 10, which demonstrate that the previous approach primarily relies on exploiting only the most common features.

---

> ### Author Response · Authors · 2024-11-25
> **Follow-Up on Review Comments**
>
> Dear Reviewer eBdC,
>
> Thank you very much for taking the time to review our paper and provide valuable feedback.
>
> We have carefully addressed all your concerns regarding writing issues and visualization. May we kindly ask if you have any further questions or suggestions to help improve our paper? We would be more than happy to address any additional feedback you may have.
>
> Thank you once again for your time and thoughtful review.

---

> > ### Comment · Reviewer_eBdC · 2024-11-26
> >
> > Thanks for the detailed rebuttal. I appreciated it and plan to increase the score to marginally above the acceptance threshold.

---

> > > ### Author Response · Authors · 2024-11-26
> > > **Thank you**
> > >
> > > Dear Reviewer eBdC,
> > >
> > > Thank you for your reply and positive comments on our work.
> > >
> > > Best regards,

---

### Official Review · Reviewer_b4Kq · 2024-11-03

**Soundness:** 3
**Presentation:** 2
**Contribution:** 3
**Rating:** 8
**Confidence:** 3

**Summary:**

This paper addresses the **incoherence problem in diffusion models**, where inconsistencies across timesteps lead to artifacts and reduced image quality. The authors introduce a novel approach by framing this issue as **feature discrepancy alignment** between predicted image distributions and representative features across timesteps. They propose **Representative Guidance (RepG)**, which uses representative vectors from self-supervised models to guide sampling coherently. This method successfully enhances the quality of generated images while preserving diversity, offering promising alternative guidance in diffusion models.

**Strengths:**

1. **Originality:**  This paper tackles the important challenge of enhancing image generation quality in diffusion models. The authors approach this by tackling incoherence in diffusion models and redefining it as a mismatch in predicted image distributions across timesteps. This fresh perspective enables a unique and novel solution to this problem.
2. **Novelty:** Unlike conventional classifier-based guidance, RepG utilizes pre-trained self-supervised models to provide representative vectors, making it effective for fine-grained image sampling. By treating sampling as a downstream task of self-supervised learning, RepG leverages the self-supervised features, which inherently contain richer semantic information than one-hot class labels. This choice addresses the common problem of classifier-guided diffusion models that often overemphasize discriminative features at the cost of generative quality and diversity.
3. **Results:** The experimental results demonstrate that RepG improves image quality, producing generations with finer details and reduced artifacts while preserving diversity.

**Weaknesses:**

1. **Dependency on Self-Supervised Model Quality**:  It seems that RepG relies heavily on the quality of self-supervised models for guidance. For instance, MoCo-v2 yields the best results, while other models like SimSiam show only marginal improvement over classifier-free guidance (e.g., FID of 1.88 vs. 1.89 on ImageNet64x64). This limitation is not thoroughly addressed in the paper, raising questions about RepG's generalizability and performance across different datasets where MoCo-style representations may not be optimal.

2. **Limited Applicability to High-Resolution Image Generation**: The paper shows promising results in lower-resolution settings, but lacks validation on high-resolution datasets. Since RepG’s main advantage lies in enriching details while preserving diversity, testing on high-resolution images could reveal strengths or limitations in its ability to maintain fine-grained coherence and detail at scale.

3. **Need for More Convincing Quantitative Results**: While Figure 3 provides examples of improvements, some results appear limited or inconsistent. For example, in the last row on the right, although a human figure is partially removed, body parts remain, which makes the result more unrealistic. Additionally, the left example in that row seems to reduce image realism rather than improve it. Including more compelling examples would strengthen the argument for RepG’s claimed improvements in detail and coherence.

**Questions:**

1. The paper suggests that RepG is more computationally efficient than classifier-based guidance, which often requires noise-aware training. However, it’s unclear what the computational trade-offs are for RepG, given the potential overhead from integrating self-supervised models and tuning features at each timestep. Could the authors provide a more thorough comparison of computational efficiency?
2. In the ablation study on the number of representative vectors 𝐾, lines 480-485, could you provide further explanation? The results suggest sensitivity to 𝐾 values. Given that the level of shared features between different classes may vary significantly across datasets and depends on the chosen self-supervised model, could the authors provide more context on how to select appropriate 𝐾 values? Additional explanations on the trade-offs involved would clarify the model's adaptability to different datasets and feature distributions.

---

> ### Author Response · Authors · 2024-11-23
> **We have addressed all of your concerns (Part 1)**
>
> We thanks the reviewer for the thorough reviews. Please find our rebuttals below:
>
> # Weaknesses discussion
>
> ## Weakness  1. SimSiam show only marginal improvement over classifier-free guidance (e.g., FID of 1.88 vs. 1.89 on ImageNet64x64)
>
>
> We cannot directly compare RepG with classifier-free guidance, as they serve distinct purposes. RepG focuses on modifying and enhancing the features within an image, while classifier-free guidance improves the conditioning of generated samples, trading off diversity to achieve better image quality and adherence to the specified conditions. In fact, RepG can be combined with classifier-free guidance to further enhance performance, as demonstrated in Table 1.
>
> Instead, in Table 2, we compare the use of RepG to sampling without it. The results demonstrate that RepG significantly enhances the performance of ADM without guidance, regardless of the choice of self-supervised models. Notable improvements are observed with MoCo v2, SimSiam, and MoCo v3, yielding scores of 1.69, 1.88, and 1.81, respectively, compared to 2.07.
>
> ## Weakness 2. MoCo-v2 yields the best results and better than Mocov3 and Simsiam.
>
> Both MoCo v3 and SimSiam eliminate the use of a negative queue during training. While this simplification aids in achieving linear separation of instance representations, it reduces the model's ability to capture information about other instances in the feature space. In contrast, MoCo v2 utilizes contrastive loss to explicitly push apart "negative features," enabling a richer and more discriminative feature space. The absence of this mechanism in MoCo v3 and SimSiam makes them less effective in our guidance method, where contrastive loss plays a critical role in supporting the generation of high-quality features.
>
> ## Weakness 3. High resolution datasets validation
>
> The computational resources limitation prevent us from providing comprehensive study on high resolution in the paper.
>
> For the rebuttals, we run a simple experiments on ImageNet512 with only 10000 samples instead of 50000 as in the paper. The results are shown in Table R1:
>
> |                            | FID   | sFID  | Prec | Recall |
> |----------------------------|-------|-------|------|--------|
> | ADM (50000 samples)*       | 23.24 | 10.19 | 0.73 | 0.60   |
> | ADM (10000 samples)        | 25.87 | 23.83 | 0.73 | 0.63   |
> | ADM + RepG (10000 samples) | 20.24 | 21.33 | 0.75 | 0.62   |
>
> Table R1: Comparison on ImageNet512x512 with RepG guidance scale=40.0, K=10. * means the values are taken from ADM paper due to the large sampling time.
>
> Even with 10000 samples, it would still take us around 2 days for running on 4 V100 GPUs. We did not have time further tune the hyper-parameters to achieve more significant results.
>
> ## Weakness 4. in the last row on the right, although a human figure is partially removed, body parts remain, which makes the result more unrealistic
>
> In fact, the left last row figure where the image removes the human body, the right figure does not show the body parts remain. Instead, if we zoom in more details, we will see that it is the hanging clothes. As a result, it is not an unrealistic generated sample.
>
> We have added more figures in Figure 11,12,13 and 14 in Appendix of the uploaded revision.

---

> ### Author Response · Authors · 2024-11-23
> **We have addressed all of your concerns (Part 2)**
>
> # Questions
>
> ## Q1. RepG is more computationally efficient than classifier-based guidance
>
> The primary computational expense of classifier guidance arises from the backpropagation through the noise-aware classifier. In contrast, representative guidance relies on much lighter models, such as ResNet-50, which significantly reduces the sampling time.
>
> Please find the computational efficiency as below tables (R2 and R3) (We use Nvidia 4090RTX for this experiment):
>
> | Model                    | Computational cost (GPU hours) |
> |--------------------------|--------------------------------|
> | No guidance              | *171.22*                         |
> | Representative guidance  | **182.36**                     |
> | Classifier guidance      | 247.84                         |
> | Classifier-free guidance | 352.89                         |
>
> Table R2. GPU hours on 1 GPU are needed to generate 50,000 images with 256x256 resolutions. Diffusion Model: ADM/ Datasets: ImageNet256x256
>
> | Model                    | Computational cost (GPU hours) |
> |--------------------------|--------------------------------|
> | No guidance              | *16.71*                        |
> | Representative guidance  | **17.55**                      |
> | Classifier guidance      | 31.52                          |
> | Classifier-free guidance | 32.64                          |
>
> Table R3. GPU hours on 1 GPU needed for generating 50000 images with 64x64 resolutions. Diffusion Model: ADM/ Datasets: ImageNet64x64
>
> ## Q2. Could the authors provide more context on how to select appropriate 𝐾 values?
> From Table 3 in the main paper, we can see with $K=5$, the performance is the best for performance for 64x64 resolution. We further evaluate the performance with different $K$ values on ImageNet256x256.
>
> Based on our observations, higher resolutions require larger values of
> $K$ to capture more detailed guidance. We conducted additional experiments on ImageNet256 using different $K$ values and found that while larger $K$ consistently improves performance for high resolutions, the outputs remain relatively stable across various $K$ values. The result is shown in Table R4.
>
> |                            | FID   | sFID  | Prec | Recall |
> |----------------------------|-------|-------|------|--------|
> | ADM (n=50000)              | 10.94 | 6.02  | 0.69 | 0.63   |
> | ADM (n=10000)              | 12.78 | 19.50 | 0.70 | 0.64   |
> | ADM + RepG (n=10000, K=1)  | 11.92 | 19.23 | 0.73 | 0.67   |
> | ADM + RepG (n=10000, K=5)  | 10.98 | 18.81 | 0.72 | 0.68   |
> | ADM + RepG (n=10000, K=10) | **10.64** | **17.51** | **0.73** | 0.67   |
> | ADM + RepG (n=10000, K=15) | 10.88 | 19.04 | 0.72 | 0.68   |
> | ADM + RepG (n=10000, K=20) | 11.85 | 19.07 | 0.72 | 0.68   |
>
> Table R4. Comparison of different $K$ values on ImageNet256x256. To save time, we do the experiments with 10000 samples instead of 50000 samples as in the paper.

---

> ### Author Response · Authors · 2024-11-25
> **Did we answer your questions?**
>
> Dear Reviewer b4Kq,
>
> We have addressed your questions and concerns regarding the writing, high-resolution details, and ablation study.
>
> Could you kindly confirm if our responses sufficiently addressed your queries? If you have any remaining concerns, we would be happy to provide further clarifications or additional details.
>
> Thank you for your time and feedback.

---

> > ### Comment · Reviewer_b4Kq · 2024-11-26
> >
> > Thank you for the detailed explanation and the additional experiments. My only concern lies still in the image generation quality of some samples.
> > Based on the motivation and novelty of this work, I will increase my score.

---

> > > ### Author Response · Authors · 2024-11-27
> > > **Thank you for your support to our works**
> > >
> > > Dear Reviewer b4Kq,
> > >
> > > Thank you for your valuable feedback and support of our work. We will continue to revise the manuscript in line with your suggestions regarding the quality of image generation.
> > >
> > >
> > > Best regards,

---

### Author Response · Authors · 2024-11-23
**We have addressed all of reviewers' concerns**

Dear Reviewers,

Thank you for your thoughtful and constructive comments.

We appreciate the effort you have put into reviewing our work. Most of the concerns raised were related to writing issues, and we have addressed them in detail for each reviewer.

We have updated the revision to incorporate your suggestions and improve clarity, consistency, and overall presentation. Please let us know if you have any further concerns or suggestions.

Thank you again for your valuable feedback.

---

### Meta-Review · Area_Chair_KciM · 2024-12-17

**Metareview:**

This paper proposes a novel guidance technique to improve sampling from generative diffusion models. Guidance is based on self-supervised representations to improve sampling and can be combined with classifier-free guidance and leads to state of the art results.
Strengths mentioned in the reviews include the novelty and originality of the approach that improves sampling (finer details, reduced artifacts) while not hurting diversity (unlike cfg), clarity of presentation, use of public datasets, good review of related work and extensive experiments.
Weaknesses include the dependence on type of self-supervised features being used, unexplored applicability in high-resolution setting, under exploration of computational costs/benefits, not clear how number of representative vectors K should be set, cluttered comparison when using ADM backbone, clarifications needed in places.

**Additional Comments On Reviewer Discussion:**

In response to the reviews the authors submitted a rebuttal and a revised manuscript. The rebuttal addressed most concerns raised by the reviewers, as acknowledges by all three reviewers. The reviewers unanimously recommend accepting the paper, and the AC follows this recommendation.

* Please note that the camera-ready page limit is identical with the submission version (10 pages).
https://iclr.cc/Conferences/2025/AuthorGuide

---

### Decision · Program_Chairs · 2025-01-22

Accept (Spotlight)